# Yeast eIF2A has a minimal role in translation initiation and uORF-mediated translational control in vivo

**Swati Gaikwad, Fardin Ghobakhlou[†, ‡], Hongen Zhang, Alan G Hinnebusch***

Division of Molecular and Cellular Biology, Eunice Kennedy Shriver National Institute of Child Health and Human Development, National Institutes of Health, Bethesda, United States

**\*For correspondence:**
ahinnebusch@nih.gov

**Present address:** †Department of Microbiology, Infectiology & Immunology, Faculty of Medicine, University of Montreal, Montreal, Canada; ‡GenoSap Inc, Montreal, Canada

**Abstract** Initiating translation of most eukaryotic mRNAs depends on recruitment of methionyl initiator tRNA (Met-tRNAi) in a ternary complex (TC) with GTP-bound eukaryotic initiation factor 2 (eIF2) to the small (40S) ribosomal subunit, forming a 43S preinitiation complex (PIC) that attaches to the mRNA and scans the 5′-untranslated region (5′ UTR) for an AUG start codon. Previous studies have implicated mammalian eIF2A in GTP-independent binding of Met-tRNAi to the 40S subunit and its recruitment to specialized mRNAs that do not require scanning, and in initiation at non-AUG start codons, when eIF2 function is attenuated by phosphorylation of its α-subunit during stress. The role of eIF2A in translation in vivo is poorly understood however, and it was unknown whether the conserved ortholog in budding yeast can functionally substitute for eIF2. We performed ribosome profiling of a yeast deletion mutant lacking eIF2A and isogenic wild-type (WT) cells in the presence or absence of eIF2α phosphorylation induced by starvation for amino acids isoleucine and valine. Whereas starvation of WT confers changes in translational efficiencies (TEs) of hundreds of mRNAs, the *eIF2AΔ* mutation conferred no significant TE reductions for any mRNAs in non-starved cells, and it reduced the TEs of only a small number of transcripts in starved cells containing phosphorylated eIF2α. We found no evidence that eliminating eIF2A altered the translation of mRNAs containing putative internal ribosome entry site (IRES) elements, or harboring uORFs initiated by AUG or near-cognate start codons, in non-starved or starved cells. Thus, very few mRNAs (possibly only one) appear to employ eIF2A for Met-tRNAi recruitment in yeast cells, even when eIF2 function is attenuated by stress.

## eLife assessment

In this **important** study, Gaikwad and colleagues employed ribosome profiling in conjunction with standard biochemical approaches to investigate the role of eIF2A in translation initiation in yeast under optimal growth conditions or stress. The authors provide **convincing** data that eIF2A is not implicated in translation initiation in yeast, a finding that is anticipated to inspire future investigations to identify the cellular role(s) of eIF2A in yeast. Considering the broad scope of cellular functions attributed to eIF2A, this study should be of interest to a wide spectrum of biomedical researchers ranging from those studying mechanisms of translation regulation to virologists and cancer biologists.

## Introduction

Eukaryotic mRNAs are generally translated by the scanning mechanism, which commences with assembly of a 43S preinitiation complex (PIC) containing the small (40S) ribosomal subunit, a ternary

complex (TC) of GTP-bound eukaryotic initiation factor 2 (eIF2) and methionyl initiator tRNA (Met-tRNAi), along with various other initiation factors. The 43S PIC attaches to the 5′-end of the mRNA, activated by eIF4F bound to the m⁷G cap (comprised of cap-binding protein eIF4E, scaffolding subunit eIF4G, and DEAD-box RNA helicase eIF4A) and scans the 5′ UTR to identify the AUG start codon, using complementarity with the anticodon of Met-tRNAi to recognize the AUG triplet. The 48S PIC arrested at the start codon joins with the large (60S) subunit to form an 80S initiation complex ready to synthesize the first peptide bond in protein synthesis (reviewed in *Hinnebusch, 2014*; *Shirokikh and Preiss, 2018*). A key mechanism for downregulating bulk translation initiation during stress entails phosphorylation of the α-subunit of eIF2, which converts eIF2-GDP from substrate to inhibitor of its guanine nucleotide exchange factor, eIF2B, diminishing TC assembly. Translation of specialized mRNAs encoding certain stress-activated transcription factors regulated by inhibitory upstream open-reading frames (uORFs) in their 5′ UTRs is induced rather than inhibited by eIF2α phosphorylation because the reduction in TC levels allows scanning PICs to bypass the uORF start codons and initiate further downstream at the start codon for the transcription factor coding sequences (CDS). This mechanism, dubbed the Integrated Stress Response in mammals, governs the translational induction of *GCN4* mRNA in budding yeast in response to amino acid starvation, dependent on the sole yeast eIF2α kinase, Gcn2. The Gcn4 protein thus induced activates the transcription of multiple genes encoding amino acid biosynthetic enzymes (reviewed in *Hinnebusch, 2005*; *Gunišová et al., 2018*; *Dever et al., 2023*).

Various studies using cultured mammalian cells have implicated the auxiliary initiation factor eIF2A in translation initiation under stress conditions where eIF2 function is reduced by phosphorylation, in non-canonical initiation events where an internal ribosome entry site (IRES) bypasses the scanning mechanism for selecting AUG codons, or when a near-cognate codon (NCC) rather than AUG serves as initiation site (reviewed in *Komar and Merrick, 2020*). eIF2A was originally characterized as a factor purified from rabbit reticulocytes that could stimulate GTP-independent recruitment of Met-tRNAi to the 40S subunit in response to an AUG codon, but could not stimulate the same reaction on native globin mRNA (*Adams et al., 1975*). Later studies suggested that the stimulation of Met-tRNAi binding was actually conferred by a protein co-purifying with eIF2A, identified as ligatin/eIF2D (*Dmitriev et al., 2010*; *Skabkin et al., 2010*), which was subsequently implicated in ribosome recycling at termination codons (reviewed in *Dever and Green, 2012*). Experiments in cultured cells suggested that eIF2A could functionally substitute for eIF2 in recruiting Met-tRNAi to the AUG start codons of viral mRNAs, including Sindbis virus 26S mRNA (*Ventoso et al., 2006*) and HCV mRNA (*Kim et al., 2011*), which is driven by an IRES, when eIF2α is phosphorylated in the virus-infected cells. Other evidence suggested that eIF2A functionally cooperates with eIF5B in recruiting Met-tRNAi to the 40S-HCV IRES PIC (*Kim et al., 2018*), and it was implicated in IRES-dependent translation of c-SRC mRNA (*Kwon et al., 2017*). However, several independent studies did not support a role for eIF2A in initiation on Sindbis or HCV mRNAs (*Jaafar et al., 2016*; *Sanz et al., 2017*; *González-Almela et al., 2018*).

Other lines of evidence implicated mammalian eIF2A in initiation at CUG initiation codons decoded by a leucyl-tRNA versus Met-tRNAi in producing polypeptides presented by major histocompatibility complex (MHC) class I molecules (*Starck et al., 2012*); and in initiation at an upstream CUG codon to produce an N-terminally extended form of PTENα (*Liang et al., 2014*). eIF2A was also reported to support non-AUG initiation events involved in translation through an expansion of GGGGGC repeats present in the C9ORF72 gene in neuronal cells exhibiting elevated eIF2α phosphorylation (*Sonobe et al., 2018*). There is additional evidence that eIF2A mediates initiation at UUG- or CUG-initiated uORFs in the 5′ UTR of BiP mRNA that appear to enhance BiP translation when eIF2α phosphorylation is induced by ER stress (*Starck et al., 2016*). eIF2A has also been implicated in stimulating translation of positive-acting uORFs initiated by CUG or other NCCs during tumor initiation in a manner important for tumor formation in animals (*Sendoel et al., 2017*).

Recently, eIF2A knock-out mice were found to exhibit reduced longevity, dysregulated lipid metabolism, obesity, decreased glucose tolerance, and increased insulin resistance—all phenotypes of metabolic syndrome—and also decreased lymphocyte production and compromised immune tolerance, implicating eIF2A in several important physiological processes and disease states. The mice lacking eIF2A, however, did not exhibit defects in expression of BiP or a protein (CHOP) upregulated by eIF2α phosphorylation during ER stress (*Anderson et al., 2021*). It is unknown whether any of the

abnormalities observed in eIF2A-deficient mice result from dysregulated translation, nor whether any of the translational functions ascribed to eIF2A in cultured mammalian cells operate in vivo.

The amino acid sequence of eIF2A is fairly well conserved between yeast and mammals across the length of the protein (*Komar and Merrick, 2020*), raising the possibility that it might function in IRES-mediated translation or non-canonical initiation at NCC codons in yeast. Analysis of a yeast mutant lacking the gene encoding eIF2A (*YGR054W*) indicated, as might be expected, that eIF2A did not participate in canonical, cap-dependent translation initiation, having no impact on bulk translation initiation in vivo. Yeast eIF2A was found, however, to co-fractionate with 40S and 80S ribosomes and to interact genetically with canonical initiation factors eIF4E and eIF5B (*Zoll et al., 2002*; *Komar et al., 2005*). Surprisingly, analysis of the deletion mutant (referred to below as *eIF2AΔ*) indicated that eIF2A functions to suppress, rather than enhance, non-canonical initiation events that occur independently of scanning, including an IRES identified in the *URE2* mRNA (*Komar et al., 2003*; *Komar et al., 2005*; *Reineke and Merrick, 2009*; reviewed in *Komar and Merrick, 2020*). It is unknown whether yeast eIF2A can substitute for eIF2 in Met-tRNAi recruitment under conditions of eIF2α phosphorylation. One piece of evidence arguing against this idea is that the *eIF2AΔ* mutation did not alter expression of a reporter for the *GCN4* transcript (*Zoll et al., 2002*) that, as mentioned above, is induced by eIF2α phosphorylation. As translational control of *GCN4* mRNA is highly specialized (*Gunišová et al., 2018*), it seemed possible that other yeast mRNAs might utilize eIF2A as an auxiliary factor for Met-tRNAi recruitment to AUG codons, or for initiation at NCCs. In this study, we explored this possibility by conducting ribosome profiling of an *eIF2AΔ* mutant in the presence or absence of increased phosphorylation of eIF2α induced by amino acid limitation. Our results do not support the possibility that eIF2A frequently substitutes for eIF2 or participates in non-canonical initiation events, even under stress conditions of attenuated eIF2 function.

## Results
### Eliminating yeast eIF2A has little impact on translational reprogramming conferred by phosphorylation of eIF2α in cells starved for amino acids

To examine whether eIF2A provides an eIF2-independent initiation mechanism for any yeast mRNAs, we first examined bulk polysome formation in a yeast mutant lacking the gene *YGR054W* encoding eIF2A (denoted *eIF2AΔ* below) and an isogenic wild-type strain (WT), both grown in nutrient-replete medium (SC) or under conditions of isoleucine/valine starvation, imposed by the drug sulfometuron methyl (SM), to induce eIF2α phosphorylation by Gcn2 and thereby reduce TC levels. We reasoned that if eIF2A can substitute for eIF2 to maintain translation initiation of a sizeable fraction of mRNAs when eIF2 function is reduced, then we might observe a depletion of polysomes in the SM-treated *eIF2AΔ* mutant compared to SM-treated WT cells. As we reported previously (*Gaikwad et al., 2021*), SM treatment of WT evoked a small reduction in the ratio of polysomes to monosomes (P/M), indicating a diminished rate of bulk translation initiation (*Figure 1A(i) and (iii)*). Essentially identical P/M ratios were observed in the corresponding untreated and SM-treated *eIF2AΔ* mutant (*Figure 1A(ii) and (iv)*), suggesting that eIF2A does not function broadly to compensate for reduced eIF2 function in the yeast translatome. These findings are consistent with a previous analysis in which eIF2A was depleted by transcriptional shut-off of an *eIF2A* allele expressed from the *GAL1* promoter (*Zoll et al., 2002*).

We next asked whether eIF2A can provide an eIF2-independent initiation mechanism for any particular mRNAs in yeast by conducting ribosome profiling of the same *eIF2AΔ* mutant and WT strains under the conditions of SM treatment employed above. Ribosome profiling entails deep-sequencing of 80S ribosome-protected mRNA fragments (RPFs, or ribosome footprints) in parallel with total RNA. The ratio of sequencing reads of RPFs summed over the CDS to the total mRNA reads for the corresponding transcript provides a measure of translational efficiency (TE) for each mRNA (*Ingolia et al., 2009*). Owing to normalization for total read number in each library, the RPF and mRNA reads and the calculated TEs are determined relative to the average values for each strain. The RPF and RNA read counts between biological replicates for each strain and condition were highly reproducible (Pearson's r ≈ 0.99) (*Figure 1—figure supplement 1A–H*).

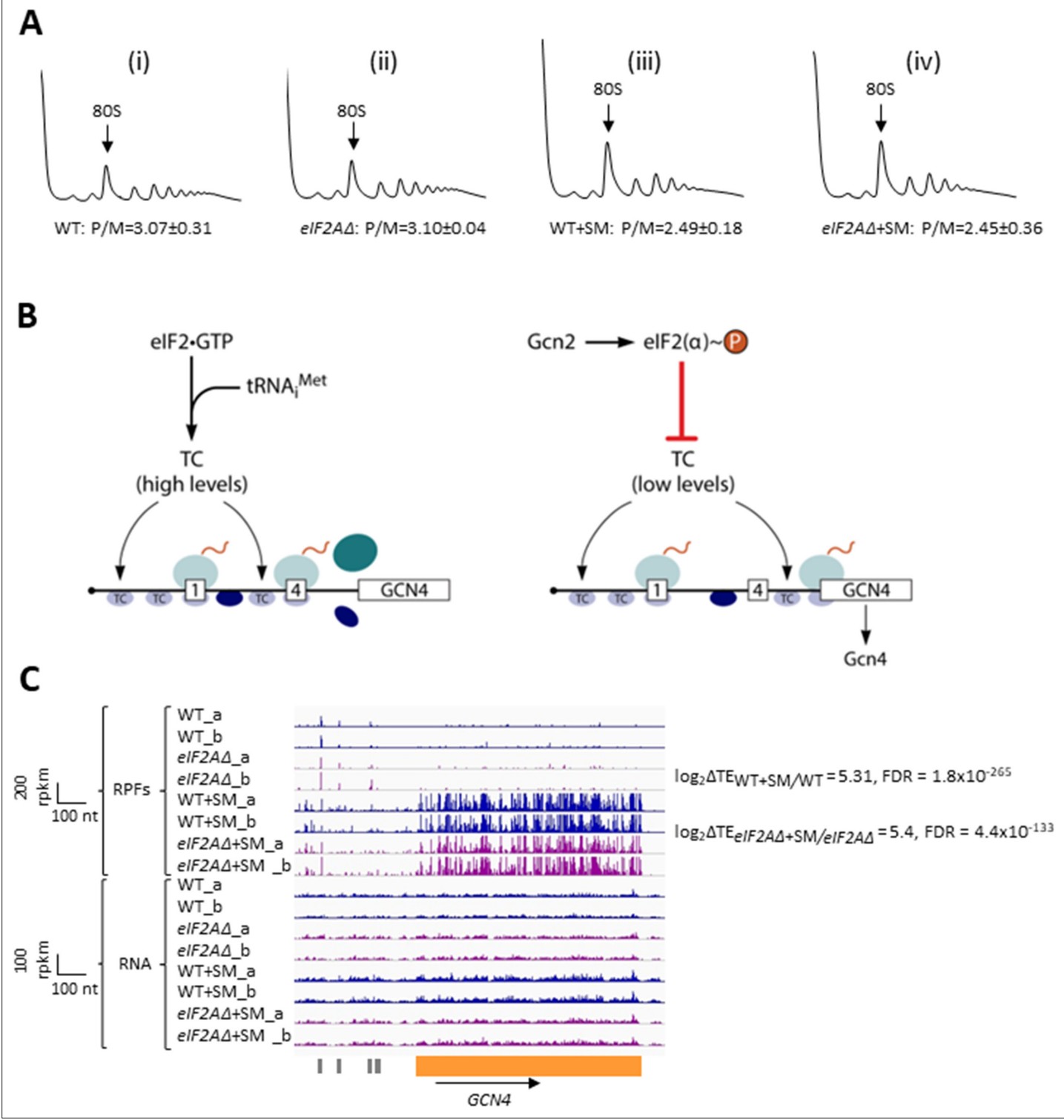

**Figure 1.** Elimination of eIF2A has no effect on bulk protein synthesis in the presence or absence of amino acid starvation. (**A**) Polysome profiles of wild-type (WT) strain (BY4741) and *eIF2AΔ* mutant (F2247) untreated (**i–ii**) or treated with sulfometuron methyl (SM) (**iii–iv**). For (**i–ii**), cells were cultured in SC medium at 30°C to log-phase and treated with 50 µg/mL of cycloheximide 5 min prior to harvesting. For (**iii–iv**), cells were cultured in SC medium lacking Ile/Val and treated with 1 µg/mL of SM for 20 min before addition of cycloheximide. Cell extracts were resolved by sedimentation through sucrose density gradients and scanned continuously at 260 nm during fractionation. The plots show the $A_{260}$ measured across the gradient with the top of the gradient on the left. (**B**) Schema of translational control of *GCN4* mRNA, wherein translation of the main coding sequences (CDS) is induced by phosphorylation of eIF2α through a specialized 'delayed reinitiation' process mediated by four short upstream open-reading frames (uORFs). (See text

*Figure 1 continued on next page*

*Figure 1 continued*

for details.) (**C**) Genome browser view of ribosome profiling data for *GCN4* mRNA. Tracks display RPF or mRNA reads mapped across the transcription unit, with the scales given in rpkm (reads per kilobase of transcript per million mapped reads). Data are presented for WT (blue) and *eIF2Δ* cells (purple) with or without SM treatment, as indicated. Each genotype/treatment includes two biological replicates, designated _a and _b. The main CDS is shown schematically in orange below the tracks and the four uORFs are in gray. The calculated values for $\log_2\Delta TE_{WT+SM/WT}$ and $\log_2\Delta TE_{eIF2AΔ+SM/eIF2AΔ+SM}$ and the respective false discovery rates (FDRs) are shown on the right.

The online version of this article includes the following figure supplement(s) for figure 1:

**Figure supplement 1.** High reproducibility between biological replicates of ribosome footprint profiling and RNA-seq analyses.

To establish that SM treatment induced comparable levels of eIF2α phosphorylation in both the WT and *eIF2AΔ* mutant, we examined the ribosome profiling data for *GCN4* mRNA, whose translation is induced by phosphorylated eIF2α by the specialized 'delayed reinitiation' mechanism imposed by the four uORFs in its transcript. Translation of the 5'-proximal uORFs (uORF1 and uORF2) gives rise to 40S subunits that escape recycling at these uORF stop codons and resume scanning downstream. At high TC levels, they quickly rebind TC and reinitiate at uORFs 3 or 4 and are efficiently recycled from the mRNA following their translation. When TC levels are reduced by eIF2α phosphorylation, a fraction of scanning 40S subunits fails to rebind TC until after bypassing uORFs 3–4, and then rebind the TC in time to reinitiate at the *GCN4* CDS, inducing *GCN4* translation (*Figure 1B*; reviewed in *Hinnebusch, 2005*; *Gunišová et al., 2018*).

Ribosome profiling revealed the expected strong induction of *GCN4* translation evoked by SM treatment of WT cells, as revealed by greatly increased RPFs in the CDS with little or no change in *GCN4* mRNA reads, yielding an increase in TE (ΔTE) of ~40-fold (*Figure 1C*, WT+SM versus WT, cf. replicate cultures _a and _b for RPFs and RNA reads). Highly similar results were observed in the *eIF2AΔ* mutant in the presence and absence of SM (*Figure 1C*, *eIF2AΔ*+SM versus *eIF2AΔ*), indicating that eIF2α phosphorylation was induced by SM at comparable levels in both the mutant and WT strains. These data argue against the possibility that eIF2A contributes to recruitment of Met-tRNA$_i$ as an alternative to the TC by 40S subunits scanning downstream from *GCN4* uORFs 1–2. If this was the case, then the *eIF2AΔ* mutation would impose a further delay in rebinding Met-tRNA$_i$ to the 40S subunits re-scanning downstream from uORFs 1–2 and thereby increase the proportion that bypass uORFs 3–4 when TCs are limiting, derepressing *GCN4* translation in SM-treated *eIF2AΔ* versus WT cells. The highly similar induction of *GCN4* translation in the two strains (*Figure 1C*) argues against such a role for eIF2A in recruiting Met-tRNAi to 40S subunits scanning the *GCN4* mRNA leader. These findings are in agreement with measurements of *GCN4-lacZ* reporter expression in SM-treated *eIF2AΔ* versus WT cells; although the induction by SM was quite modest in that study, and an auxiliary role for eIF2A might have gone undetected (*Zoll et al., 2002*).

We turned next to the question of whether eliminating eIF2A alters the translation of any other yeast mRNAs, reasoning that mRNAs able to utilize eIF2A in place of TC for recruitment of Met-tRNA$_i$ would exhibit greater TE reductions evoked by SM in *eIF2AΔ* mutant versus WT cells. If such mRNAs could be translated efficiently utilizing TC alone, then they would require eIF2A conditionally, that is, only when eIF2 function is reduced during starvation. If instead such mRNAs rely primarily on eIF2A for efficient initiation regardless of TC levels, they would exhibit reduced TEs in *eIF2AΔ* versus WT cells in the absence of SM treatment. Importantly, DESeq2 analysis of the ribosome profiling data obtained for the *eIF2AΔ* mutant and WT strains cultured in the absence of SM revealed no significant TE reductions in the mutant, as no mRNAs exhibited $TE_{eIF2AΔ}/TE_{WT}$ ratios < 1 at a false discovery rate (FDR < 0.25) that is appropriate for two highly correlated biological replicates (*Lamarre et al., 2018*; *Figure 2A*). This finding suggests that few, if any, mRNAs are appreciably dependent on eIF2A for translation initiation in nutrient-replete cells where TC is abundant.

In addition to induction of *GCN4* translation, SM treatment of WT cells leads to a broad reprogramming of translational efficiencies as DESeq2 analysis revealed hundreds of mRNAs exhibiting increases or decreases in relative TE even at the highly stringent FDR of < 0.01 (*Figure 2B*). Previously, we presented evidence that many of these TE changes conform to a pattern in which mRNAs that are efficiently translated in untreated WT cells tend to exhibit increased relative TEs, whereas poorly translated mRNAs tend to show reduced relative TEs, when phosphorylation of eIF2α is induced by SM. This same pattern was evident under two other conditions in which 43S PIC assembly is reduced, impaired recycling of 40S subunits from post-termination complexes at stop codons and depletion of

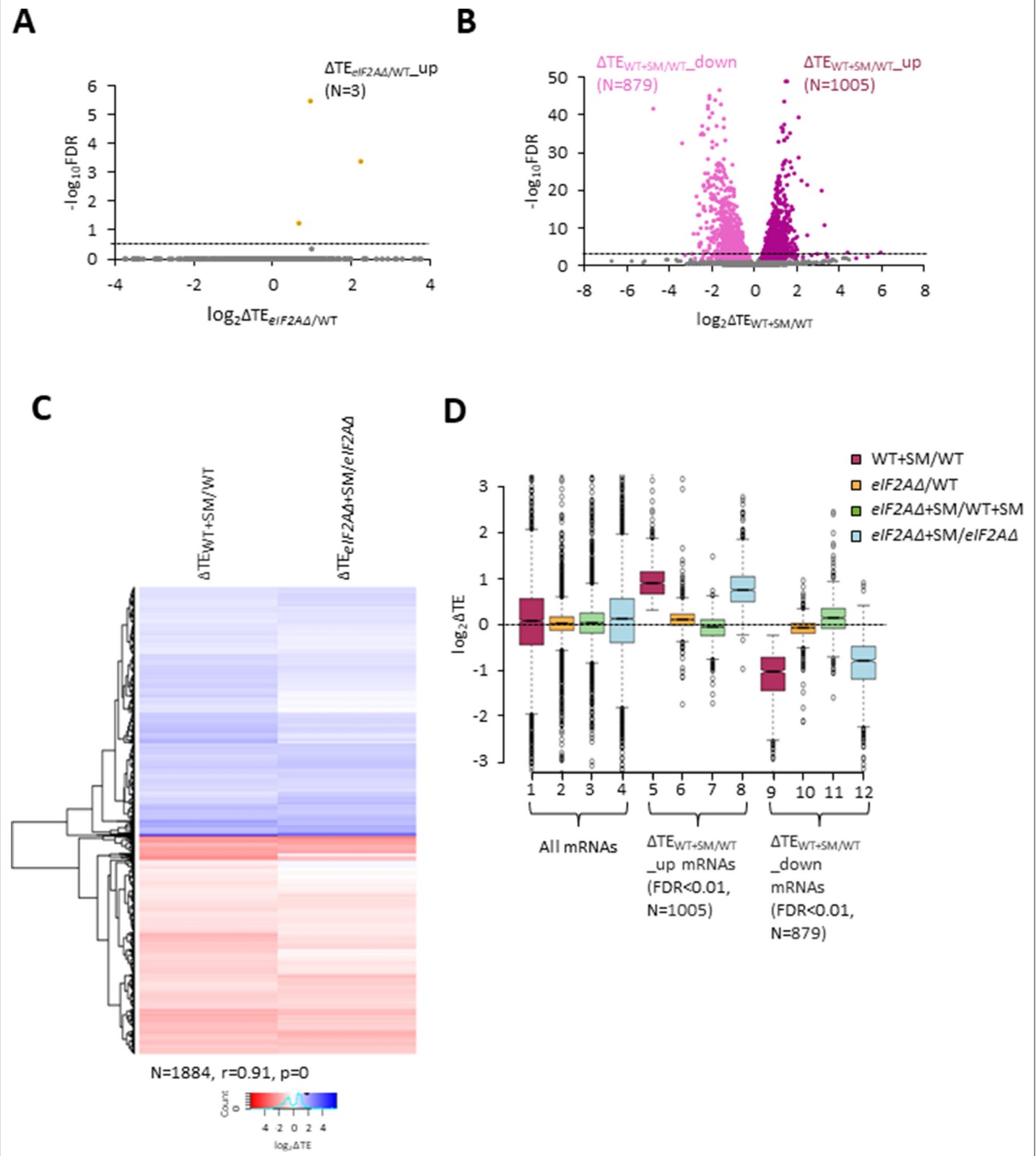

**Figure 2.** eIF2A is not critical for translation of any individual mRNAs in non-starved cells and has little impact on the reprogramming of translational efficiencies (TEs) conferred by amino acid starvation. (**A**) Volcano plot depicting the log$_2$ ratios of TEs in *eIF2AΔ* versus wild-type (WT) cells (log$_2$ΔTE$_{eIF2AΔ/WT}$ values) for each mRNA (x-axis) versus negative log$_{10}$ of the false discovery rate (FDR) (y-axis) determined by DESeq2 analysis of ribosome profiling data for the 5340 mRNAs with evidence of translation. Genes showing a significant increase in TE in *eIF2AΔ* versus WT cells at FDR < 0.25 (ΔTE$_{eIF2AΔ+SM/}$

*Figure 2 continued on next page*

*Figure 2 continued*

WT_up) are plotted in orange circles. The dotted line marks the 25% FDR threshold, below which all other 5337 mRNAs are plotted in gray. (**B**) Volcano plot as in (**A**) showing the $\log_2$ ratios of TEs in WT+ sulfometuron methyl (SM) cells versus WT cells ($\log_2\Delta TE_{WT+SM/WT}$ values) for the 5441 mRNAs with evidence of translation. The dotted line marks the 1% FDR threshold. Genes showing a significant increase ($\Delta TE_{eIF2A\Delta+SM/WT\_up}$) or decrease ($\Delta TE_{WT+SM/WT\_down}$) in TE in WT+SM versus WT cells at FDR < 0.01 are plotted in magenta and pink circles, respectively. (**C**) Hierarchical clustering analysis of $\log_2\Delta TE$ values for the 1884 mRNAs (arrayed from top to bottom) that exhibit significant TE decreases or increases in SM-treated versus untreated WT cells at FDR < 0.01 (defined in (**B**)) conferred by SM treatment of WT cells (column 1) or SM treatment of *eIF2A\Delta* cells (column 2), with the $\log_2\Delta TE$ values represented on a color scale ranging from 4 (dark blue) to –4 (dark red). The Pearson coefficient (r) and corresponding p-value for the correlation between $\log_2\Delta TE$ values in the two columns are indicated below. (**D**) Notched box plots of $\log_2\Delta TE$ values for the indicated mutant/condition for all mRNAs (columns 1–4) or for the indicated mRNA groups identified in (**B**). The y-axis scale was expanded by excluding a few outliers from the plots.

The online version of this article includes the following source data and figure supplement(s) for figure 2:

**Source data 1.** Spreadsheet tabulates the $\log_2$ ratios of translational efficiencies (TEs) in *eIF2A\Delta* versus wild-type (WT) cells ($\log_2\Delta TE_{eIF2A\Delta/WT}$ values) for each mRNA and the corresponding false discovery rate (FDR) determined by DESeq2 analysis of ribosome profiling and parallel RNA-Seq data for the 5340 mRNAs with evidence of translation (***Figure 2A***).

**Source data 2.** Spreadsheet tabulates the $\log_2$ ratios of translational efficiencies (TEs) in SM treated versus untreated WT cells ($\log_2\Delta TE_{WT+SM/WT}$ values) and the corresponding false discovery rate (FDR) determined by DESeq2 analysis of ribosome profiling and parallel RNA-Seq data for the 5441 mRNAs with evidence of translation (***Figure 2***, ***Figure 2—figure supplement 1***).

**Source data 3.** Spreadsheet tabulates the $\log_2$ ratios $\log_2\Delta TE$ values for the indicated mutant/condition for the indicated mRNA groups identified in ***Figure 2B and D***.

**Figure supplement 1.** Relative translational efficiency (TE) changes evoked by increased eIF2α phosphorylation in cells lacking eIF2A are broadly similar to relative TE changes conferred by increased eIF2α phosphorylation in wild-type (WT) cells.

**Figure supplement 1—source data 1.** Spreadsheet tabulates the $\log_2$ ratios of translational efficiencies (TEs) in sulfometuron methyl (SM)-treated *eIF2A\Delta* versus untreated *eIF2A\Delta* cells ($\Delta TE_{eIF2A\Delta+SM/eIF2A\Delta}$ values) for each mRNA and the corresponding false discovery rate (FDR) determined by DESeq2 analysis of ribosome profiling and parallel RNA-Seq data for the 5426 mRNAs with evidence of translation (***Figure 2—figure supplement 1***).

an essential 40S ribosomal protein. We proposed that this stereotypical reprogramming of translation arises from increased competition among mRNAs for limiting PICs wherein strongly translated mRNAs outcompete weakly translated mRNAs to skew TE increases towards 'strong' mRNAs (***Gaikwad et al., 2021***). SM treatment of the *eIF2A\Delta* mutant also produced a broad reprogramming of TEs involving hundreds of mRNAs translated relatively better or worse on SM treatment (***Figure 2—figure supplement 1A***). Comparing the TE changes conferred by SM in WT versus *eIF2A\Delta* cells for the mRNAs showing TE changes in WT revealed that the majority of transcripts showed TE changes in the same direction on SM treatment of WT and *eIF2A\Delta* cells (***Figure 2C***). Indeed, the majority of mRNAs dysregulated in the *eIF2A\Delta* mutant also showed TE changes in the WT strain (***Figure 2C***, ***Figure 2—figure supplement 1B***) and a very strong positive correlation exists, with a coefficient of 0.91, between the TE changes conferred by SM in the two strains. Thus, elimination of eIF2A did not substantially alter the global reprogramming of translation produced by phosphorylation of eIF2α.

If certain mRNAs depend more heavily on eIF2A for Met-tRNAi recruitment when eIF2α is phosphorylated, we might expect to observe additive reductions in TE when combining elimination of eIF2A by the *eIF2A\Delta* mutation with inhibition of eIF2 by SM treatment. Interrogating the large group of 879 mRNAs that showed TE reductions on SM treatment of WT cells revealed that they show a relatively smaller, not larger, decrease in median TE on SM treatment of the *eIF2A\Delta* mutant versus SM treatment of WT (***Figure 2D***, column 12 versus column 9). Consistent with this finding, most of these mRNAs exhibit a small increase in TE on comparing SM-treated *eIF2A\Delta* to SM-treated WT cells (***Figure 2D***, column 11). (In these and all other box plots, when the notches of different boxes do not overlap, their median values are judged to differ significantly with a 95% confidence level. As shown in columns 1–4, the median TE change for all ~5500 expressed mRNAs detected in our profiling experiments is close to unity ($\log_2 = 0$) for all of the comparisons examined in ***Figure 2D***). These results suggest that the majority of mRNAs whose translation is diminished by eIF2α phosphorylation in WT cells do not utilize eIF2A for a back-up initiation mechanism that would mitigate their TE reductions when eIF2 is impaired. Our finding that most mRNAs exhibit somewhat greater TEs in *eIF2A\Delta* versus WT cells when both are treated with SM (***Figure 2D***, column 11) might indicate that eIF2A generally acts to repress the translation of these mRNAs rather than augmenting eIF2 function in Met-tRNAi recruitment. This would not be the case in the absence of SM, however, as the results in column 10 of ***Figure 2D*** suggest a small positive effect of eIF2A on translation of these mRNAs in non-starved cells.

## Only a few mRNAs exhibit translational reprogramming consistent with eIF2A functioning as a back-up to eIF2

To determine whether there are any individual mRNAs that show greater TE reductions in response to SM treatment when eIF2A is absent, we conducted DESeq2 analysis of the TE changes in SM-treated *eIF2AΔ* versus SM-treated WT cells. A group of only 32 mRNAs showed significant TE reductions in this comparison, that is, $TE_{eIF2AΔ+SM}/TE_{WT+SM} < 1$, FDR < 0.25 (*Figure 3A*). The TE reductions for this group of transcripts (designated $\Delta TE_{eIF2AΔ+SM/WT+SM\_down}$) in comparison to all mRNAs were nearly two-fold greater in SM-treated *eIF2AΔ* versus SM-treated WT cells (*Figure 3B*, column 1), which are the results expected if they utilize eIF2A as a back-up when eIF2 is impaired by phosphorylation. They also showed TE reductions in SM-treated versus untreated *eIF2AΔ* cells, albeit of lesser magnitude (*Figure 3B*, column 2), also in the manner expected if eIF2 and eIF2A play redundant roles in their translation. Functional redundancy is further supported by the findings that both SM treatment of WT cells and elimination of eIF2A from untreated cells does not reduce their median TEs (*Figure 3B*, columns 3–4), as only one of the two factors is impaired or eliminated in these latter comparisons. The fact that the median TE of these mRNAs increases rather than decreases on SM treatment of WT (*Figure 3B*, column 3) might be explained by proposing that their ability to rely on eIF2A provides them with a competitive advantage with mRNAs that depend solely on eIF2 when the latter is impaired by phosphorylation.

To determine how many of these 32 transcripts exhibit TE reductions exclusively when both eIF2 and eIF2A are impaired/eliminated, we conducted hierarchical clustering of the TE changes in the four comparisons described above, displaying the magnitude of changes with a heat map. Only 17 of the 32 transcripts (marked with '#') displayed the diagnostic pattern of an appreciable reduction in TE both on elimination of eIF2A in SM-treated cells and on SM treatment of cells lacking eIF2A (red or pink hues in columns 1–2 of *Figure 3C*), but either a lesser reduction, no change, or increase in TE on SM treatment of WT cells and on elimination of eIF2A from untreated cells (light pink, white or blue hues in columns 3–4 of *Figure 3C*).

In an effort to provide independent evidence that a subset of the transcripts analyzed in *Figure 3C* are dependent on eIF2A only when eIF2 function is diminished by SM, we constructed luciferase reporters for 8 of the aforementioned 17 genes (*CHS5, DBF4, DNA2, HKR1, MET4, PBP4, PUS7, RAD9*) and for 4 other genes that satisfied only one of the two criteria for conditional dependence on eIF2A when eIF2 is impaired by phosphorylation (*SAG1, SVL3, NET1,* and *NSI1*). The promoters and 5′ UTR sequences of the candidate genes were fused to the firefly luciferase CDS and the 3′ UTR and transcription termination sequences of the yeast *RPL41A* mRNA and introduced into yeast on single-copy plasmids (*Sen et al., 2015*). Assaying these *LUC* reporters in both WT and *eIF2AΔ* cells in the presence and absence of SM showed that only the *HKR1* reporter displayed a significantly greater reduction in expression in response to SM treatment in *eIF2AΔ* versus WT cells (*Figure 3D*). Hence, except for *HKR1* mRNA, the *LUC* reporter analysis failed to support a conditional requirement for eIF2A for the candidate genes examined.

As an independent approach, we determined the distributions of native candidate mRNAs that co-sedimented with 80S monosomes or different polysomal species in cell extracts, examining *HKR1* and two other mRNAs (*CHS5* and *DBF4*) that satisfied both of the aforementioned criteria for conditional dependence on eIF2A, three mRNAs (*NET1, SAG1, SVL3*) that satisfied only one of the two criteria, and *ACT1* and *TRM44* mRNAs examined as negative controls. The amounts of each mRNA found in monosome or polysomal fractions were multiplied by the number of ribosomes per mRNA in that fraction (one for monosomes, two for disomes, three for trisomes, etc.) to calculate the abundance of ribosomes translating the mRNA and normalized to the input level of the mRNA in the unfractionated extracts to calculate the TE of the transcript in each condition. Analyzing the results from three biological replicates for WT versus *eIF2AΔ* cells ± SM revealed that only three transcripts, *HKR1, SAG1,* and *SVL3*, displayed a greater TE reduction in response to SM treatment of *eIF2AΔ* cells compared to SM treatment of WT (*Figure 3, Figure 3—figure supplement 1*). Thus, only the *HKR1* transcript satisfied all of the criteria for conditional eIF2A dependence in ribosome profiling data in a manner confirmed by both reporter analysis and polysome profiling. We conclude that there are very few mRNAs, possibly only one, that utilize eIF2A as a back-up for eIF2 in recruitment of tRNAi when eIF2 is impaired by phosphorylation.

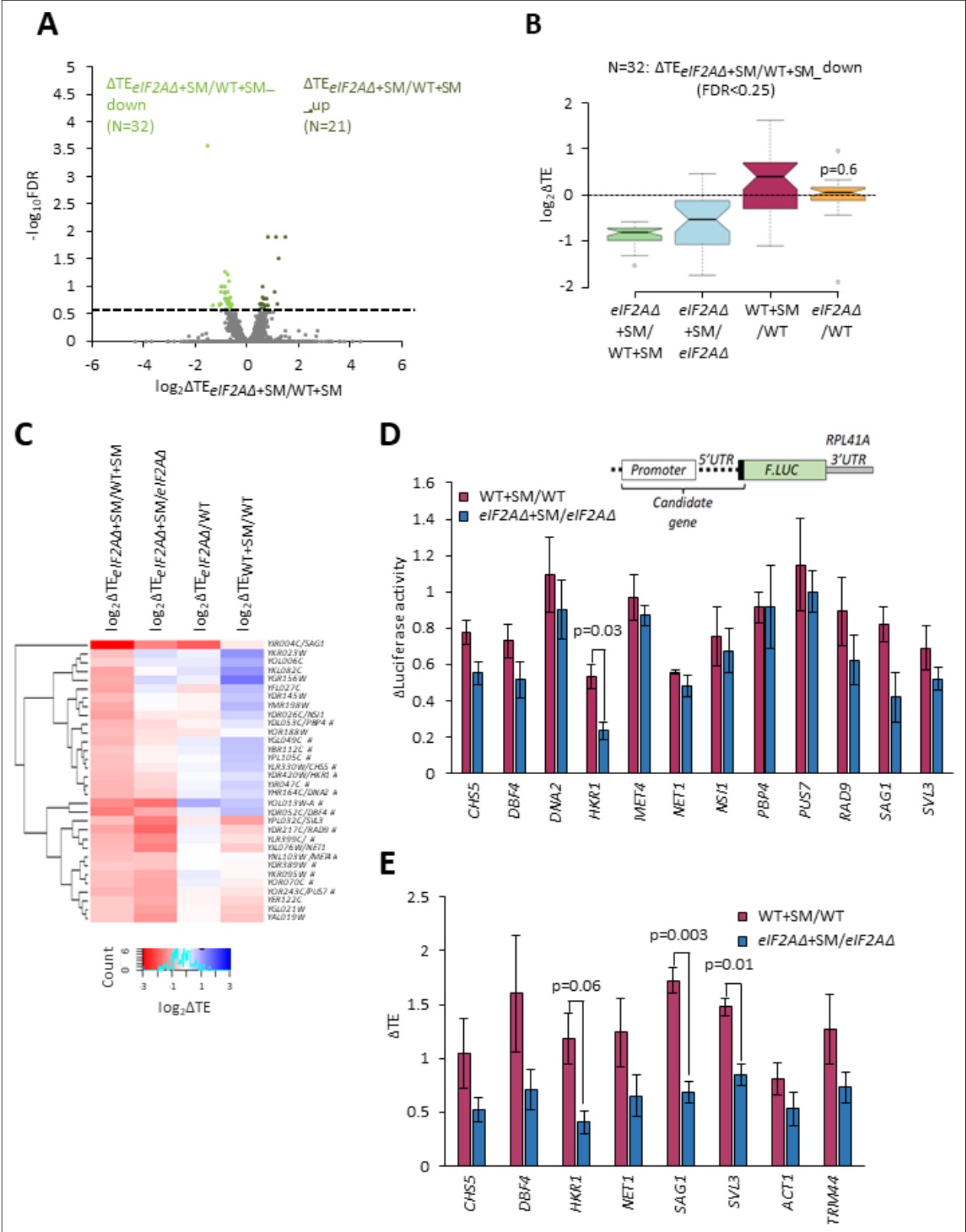

**Figure 3.** Examination of a small group of mRNAs showing evidence of a conditional requirement for eIF2A when eIF2 is impaired. (**A**) Volcano plot as in *Figure 2A* showing the log$_2$ ratios of translational efficiencies (TEs) in *eIF2A$\Delta$* cells treated with sulfometuron methyl (SM) versus wild-type (WT) cells treated with SM (log$_2\Delta$TE$_{eIF2A\Delta+SM/WT+SM}$ values) for the 5482 mRNAs with evidence of translation. The dotted line marks the 25% false discovery rate (FDR) threshold. Genes exhibiting a significant increase ($\Delta$TE$_{eIF2A\Delta+SM/WT+SM\_up}$) or decrease ($\Delta$TE$_{eIF2A\Delta+SM/WT+SM\_down}$) at FDR < 0.25 are plotted in dark or light green

*Figure 3 continued on next page*

*Figure 3 continued*

circles, respectively. (**B**) Notched box plots of log$_2$ΔTE values for the indicated mutant/condition for the 32 mRNAs in the group ΔTE$_{eIF2AΔ+SM/WT+SM\_down}$ defined in (**A**). The y-axis scale was expanded by excluding a few outliers from the plots. Statistical significance determined using the Mann–Whitney *U* test is indicated for the changes in column 4 compared to the changes observed for all mRNAs. (**C**) Hierarchical clustering analysis of log$_2$ΔTE values for the 32 mRNAs (arrayed from top to bottom) in the group defined in (**A**) for the four comparisons listed across the top, with log$_2$ΔTE values represented on a color scale ranging from 4 (dark blue) to –4 (dark red). The systematic gene names are listed for all 32 mRNAs, and the common name is indicated for those genes subjected to *LUC* reporter analysis below. Genes marked with '#'s display the pattern of TE changes consistent with conditional stimulation by eIF2A when eIF2 function is reduced by phosphorylation. Only 17 of the 32 transcripts (marked with '#') displayed the diagnostic pattern of an appreciable reduction in TE both on elimination of eIF2A in SM-treated cells and on SM treatment of cells lacking eIF2A (red or pink hues in columns 1–2) but either a lesser reduction, no change, or increase in TE on SM treatment of WT cells and on elimination of eIF2A from untreated cells (light pink, white or blue hues in columns 3–4). (**D**) Expression of *LUC* reporters in different strains/conditions constructed for selected candidate genes analyzed in (**C**). The schematic depicts reporter construct design wherein the native gene promoter, 5' UTR, and first 20 codons of the coding sequences (CDS) are fused to firefly luciferase coding sequences (*F.LUC*), followed by a modified *RPL41A* 3' UTR. Plasmid-borne reporter constructs were introduced into the WT and *eIF2AΔ* strains and three independent transformants were cultured in SC-Ura medium at 30°C to log phase (-SM) or treated with SM at 1 µg/mL after log-phase growth in SC-Ura/Ile/Val and cultured for an additional 6 hr before harvesting. Luciferase activities were quantified in whole-cell extracts (WCEs), normalized to total protein, and reported as fold change in relative light units (RLUs) per mg of protein, as means (± SEM) determined from the replicate transformants. The changes in luciferase activity plotted for each of the two comparisons depicted in the histogram were calculated as ratios of the appropriate mean activities. Results of Student's *t*-tests of the differences in fold changes between the indicated mutations/ conditions are indicated. (**E**) Determination of relative TEs for the native mRNAs of selected candidate genes analyzed in (**C, D**). Cells were cultured in the four conditions described in (**D**), WCEs were resolved by sedimentation through 10–50% sucrose gradients, and fractions were collected while scanning at 260 nm. Total RNA was extracted from 80S and polysome fractions, and the abundance of each target mRNA was quantified in each fraction by qRT-PCR, and normalized for (i) the amounts of 18S rRNA quantified for the same fractions and (ii) for the total amounts of monosomes/polysomes recovered in the gradient. The resulting normalized amounts of mRNA in each fraction were multiplied by the number of ribosomes per mRNA in that fraction, summed across all fractions, and divided by the input amount of mRNA in the WCEs, normalized to *ACT1* mRNA, to yield the TE for that mRNA in each condition. (See 'Materials and methods' for further details.) The changes in TE conferred by SM treatment of WT or *eIF2AΔ* cells were calculated for each replicate culture, untreated or SM-treated, and the mean TE changes with standard error of the means (SEMs) were plotted for the indicated comparisons. The results of Student's *t*-tests of the differences in mean TE changes are indicated.

The online version of this article includes the following source data and figure supplement(s) for figure 3:

**Source data 1.** Spreadsheet tabulates the log$_2$ ratios of translational efficiencies (TEs) in *eIF2AΔ* cells treated with sulfometuron methyl (SM) versus wild-type (WT) cells treated with SM (log$_2$ΔTE$_{eIF2AΔ+SM/WT+SM}$ values) for each mRNA and the corresponding false discovery rate (FDR) determined by DESeq2 analysis of ribosome profiling and parallel RNA-Seq data for the 5482 mRNAs with evidence of translation (*Figure 3A*).

**Source data 2.** Spreadsheet tabulates the log$_2$ΔTE values for the indicated mutant/condition for the 32 mRNAs in the group ΔTE$_{eIF2AΔ+SM/WT+SM\_down}$ defined in *Figure 3A and B*.

**Source data 3.** Spreadsheet tabulates the log$_2$ΔTE values for the indicated mutant/condition for the 32 mRNAs in the group ΔTE$_{eIF2AΔ+SM/WT+SM\_down}$ defined in *Figure 3A and C*.

**Source data 4.** Spreadsheet tabulates the changes in luciferase activity expressed from *F.LUC* reporters calculated for sulfometuron methyl (SM)-treated versus untreated wild-type (WT) and SM-treated versus untreated *eIF2AΔ* for three biological replicates (*Figure 3D*).

**Source data 5.**

**Figure supplement 1.** Representative separation of polysomes by sedimentation through a sucrose density gradient in the experiment depicted in *Figure 3E*.

## Eliminating yeast eIF2A has little consequence on translation of mRNAs harboring IRESs or upstream open-reading-frames

We next interrogated our ribosome profiling data for the effects of deleting *eIF2A* on translation of *URE2* mRNA, reported to contain an IRES that is inhibited by eIF2A (*Komar et al., 2005*). We found no significant change in TE or ribosome occupancy for *URE2* mRNA on deletion of *eIF2A* in either non-starved or SM-starved cells (*Figure 4A*). It was reported that yeast mRNAs *GIC1* and *PAB1* also contain IRESs and are subject to translational repression by eIF2A (*Reineke and Merrick, 2009*); however, we again found no significant alteration in their TEs between *eIF2AΔ* and WT cells in the presence or absence of SM treatment (*Figure 4B and C*). Our results do not support a role for eIF2A in repressing IRES function in yeast cells. It should be noted, however, that eliminating eIF2A would not be expected to confer a marked increase in TE if the IRESs make only a small contribution to overall translation of these mRNAs under the conditions of our experiments.

It has been reported that mammalian eIF2A is required for the translation of certain non-AUG initiated uORFs and that translation of these uORFs upregulates translation of the downstream CDS when eIF2α is phosphorylated during ER stress (*Starck et al., 2016*) or during tumor initiation (*Sendoel*

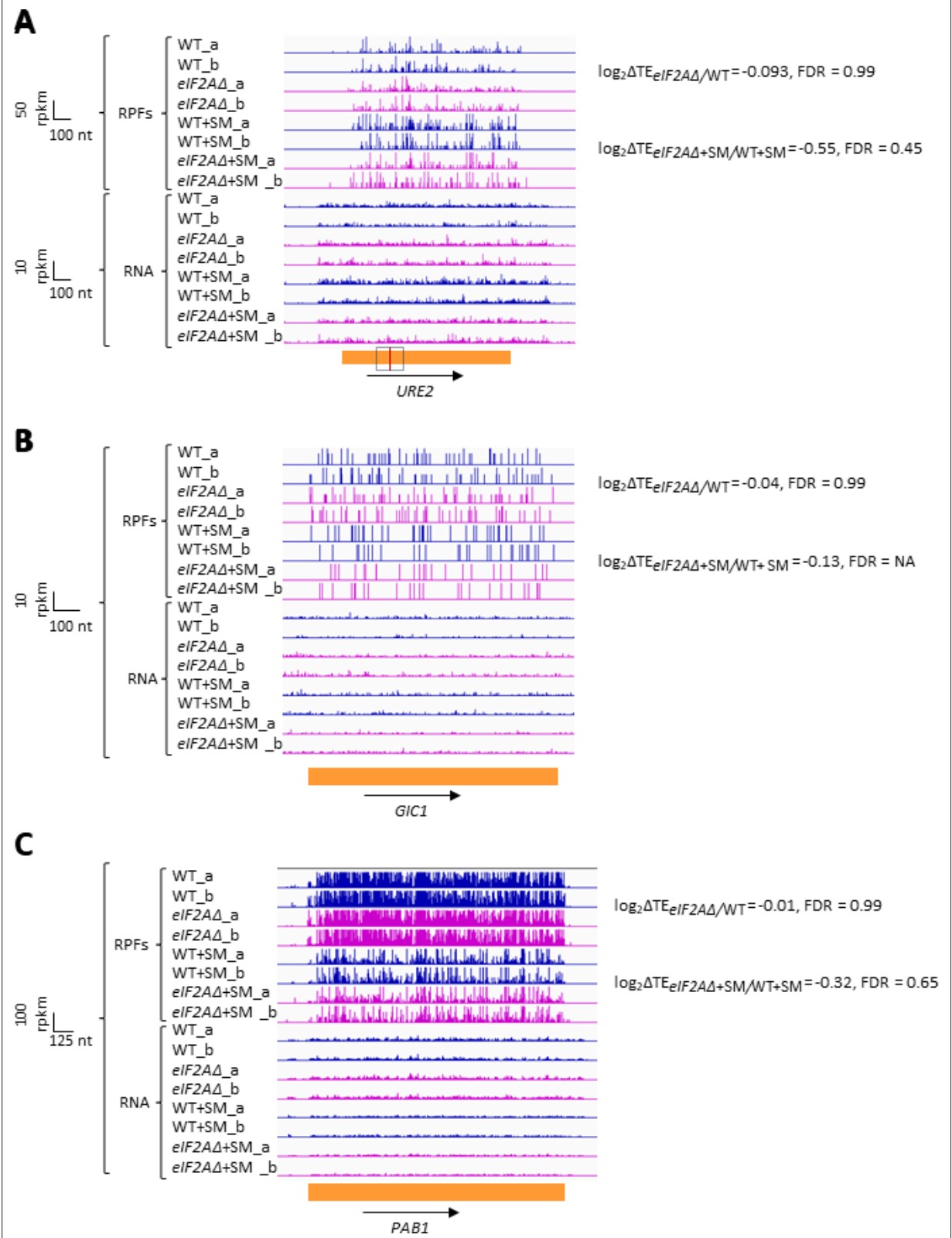

**Figure 4.** eIF2A has little or no effect on the translation of three mRNAs reported to contain internal ribosome entry sites (IRESs). Genome browser views of RPF and RNA reads from ribosome profiling data for (**A**) *URE2* mRNA, (**B**) *GIC1* mRNA, and (**C**) *PAB1* mRNA presented as in *Figure 1C*. The calculated values for $\log_2\Delta TE_{eIF2A\Delta/WT}$ and $\log_2\Delta TE_{eIF2A\Delta+SM/WT+SM}$ with the respective false discovery rates (FDRs) are shown on the right. The scale is shown on the left. The region containing the *URE2* IRES is enclosed in a dotted box, with the AUG start codon highlighted in red. Locations of the *GIC1* and *PAB1* IRESs have not been defined.

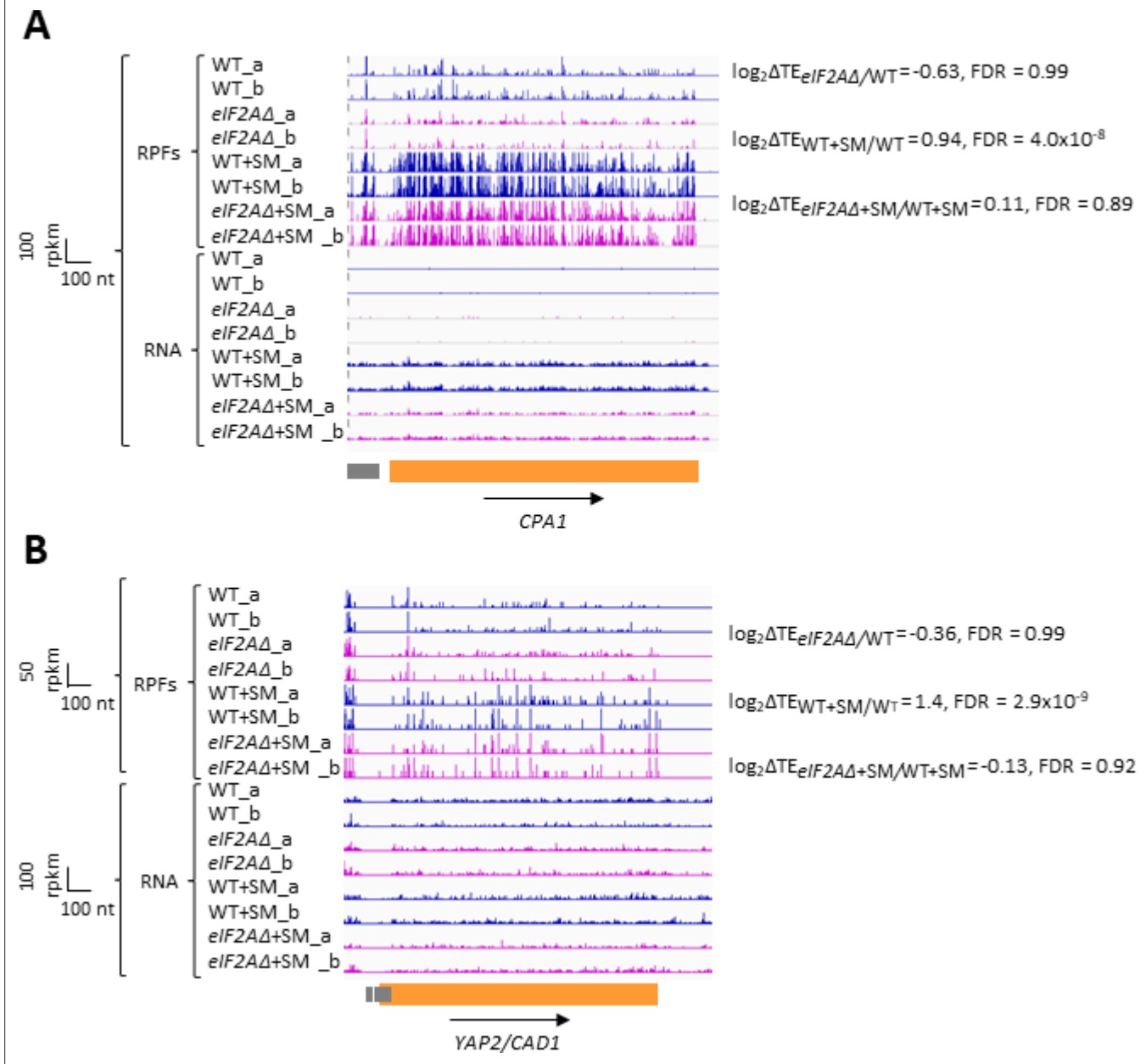

**Figure 5.** eIF2A plays little or no role in upstream open-reading frame (uORF)-mediated translational control of *CPA1* or *YAP2/CAD1* mRNA. Genome browser views of RPF and RNA reads from ribosome profiling data for (**A**) *CPA1* mRNA and (**B**) *YAP2/CAD1* mRNA presented as in *Figures 1 and 4*. Coding sequences (CDS) and uORFs are represented in orange and gray rectangles, respectively.

*et al., 2017*). Hence, we examined our profiling data more closely to determine if eIF2A might affect the translation of mRNAs harboring uORFs in yeast cells. We did not restrict our attention to stimulatory uORFs, which are rare in yeast (*May et al., 2023*), but included all uORFs in our analyses.

In addition to *GCN4* described above, *CPA1* mRNA, encoding an arginine biosynthetic enzyme, contains a single AUG-initiated inhibitory uORF that attenuates translation of the main CDS in cells replete with arginine (*Werner et al., 1987*; *Delbecq et al., 1994*). In accordance with previous findings that the inhibitory effects of single uORFs can be diminished by eIF2α phosphorylation by allowing scanning ribosomes to bypass the uORF start codon (*Young and Wek, 2016*; *Dever et al., 2023*), we observed an approximately two-fold increase in the TE of *CPA1* mRNA on SM treatment of WT cells (*Figure 5A*). However, this TE change was not significantly altered by the *eIF2AΔ* mutation. Similarly, for the well-characterized *YAP2/CAD1* mRNA, containing two inhibitory AUG uORFs (*Vilela et al.,*

*1998*), we observed a 2.6-fold TE increase conferred by SM in WT cells that, again, was not significantly altered by deletion of *eIF2A* (*Figure 5B*). The *eIF2AΔ* deletion also had no effect on the TEs of *CPA1* and *YAP2/CAD1* in non-starved cells. Thus, although we found evidence that translational repression exerted by the uORFs in *CPA1* and *YAP2/CAD1* mRNAs is mitigated by eIF2α phosphorylation, it appears that eIF2A has little or no role in recognizing the AUG codons of these uORFs in the presence or absence of diminished TC levels, just as we concluded above for the *GCN4* uORFs.

Looking more broadly, we interrogated a previously identified group of 1306 mRNAs containing 2720 uORFs, initiating with either AUG or one of the nine NCCs, that showed evidence of translation in multiple ribosome profiling datasets from various mutant and WT strains (*Zhou et al., 2020*). We also analyzed a second set of 791 mRNAs containing 982 AUG- or NCC-initiated uORFs that are both evolutionarily conserved and show evidence of translation in ribosome profiling experiments (*Spealman et al., 2018*). These four groups of mRNAs with translated AUG- or NCC-uORFs showed little or no TE change for their CDSs on SM treatment of WT cells (*Figure 6A(i)–(ii)*, columns 4 and 7 versus 1), indicating that, in contrast to *GCN4, CPA1,* and *YAP2/CAD1* mRNAs, most of them do not contain inhibitory uORFs that can be bypassed in response to eIF2α phosphorylation. Both groups of mRNAs containing either AUG- or NCC-initiated uORFs exhibit very small, albeit highly significant, reductions in CDS TEs in the *eIF2AΔ* mutant versus WT cells in the absence of SM (*Figure 6A(i)–(ii)*, columns 5 and 8 versus 2); however, the same was not observed for the effects of *eIF2AΔ* in SM-treated cells where eIF2 function is attenuated (*Figure 6(i)–(ii)*, columns 6 and 9 versus 3).

Evidence of translation of a uORF does not guarantee that it has an appreciable impact on the proportion of scanning ribosomes that reach the downstream CDS. Recently, groups of 557 mRNAs containing AUG-initiated uORFs and 191 mRNAs with NCC-uORFs were identified in which the uORFs were shown to influence translation of downstream CDSs by massively parallel analysis of reporters containing the native 5′ UTRs in comparison to mutant reporters lacking the uORF start codons (*May et al., 2023*). It was reported that among the 407 mRNAs containing a single functional AUG-uORF wherein mutating the uORF significantly altered reporter expression by >1.5-fold, all but 13 of the uORFs functioned to inhibit reporter expression. Among the 144 mRNAs containing a single functional NCC-uORF, only 13 of the uORFs influenced reporter expression either positively or negatively by >1.5-fold, reflecting the weaker effects of NCC- versus AUG-initiated uORFs on downstream translation (*May et al., 2023*). Our analysis of the 394 mRNAs containing a single inhibitory AUG-uORF— the only group large enough for statistical analysis of TE changes—revealed little or no change in median TE in response to SM treatment or to the *eIF2AΔ* mutation in the presence or absence of SM (*Figure 6A(iii)*, columns, 4–6 versus 1–3). These findings suggest that eIF2A has a minimal role in determining whether functional inhibitory AUG-initiated uORFs are translated or bypassed by scanning ribosomes in the presence or absence of eIF2α phosphorylation. These results support our findings above that the majority of mRNAs containing translated uORFs show little response to either eIF2α phosphorylation, the elimination of eIF2A, or a combination of both perturbations.

We probed more deeply into the three sets of uORF-containing mRNAs mentioned above by examining the subsets of each group that exhibit >1.41-fold increases in TE in response to SM treatment of WT cells, making them novel candidates for mRNAs controlled by inhibitory uORFs that can be overcome by eIF2α phosphorylation. These uORF-containing mRNAs exhibit median TE increases of approximately two-fold on SM treatment of WT cells (*Figure 6B(i)–(iii)*, maroon data), similar to the findings above for *CPA1* and *CAD1* mRNAs (*Figure 5A and B*). Unlike the larger groups of uORF-containing mRNAs from which they derive (analyzed in *Figure 6A(i)–(iii)*), these subsets showing appreciable TE induction by SM exhibit slightly reduced median TEs in response to the *eIF2AΔ* mutation in SM-treated, but not untreated cells (*Figure 6B(i)–(iii)*, green versus orange data). One way to explain these findings would be to hypothesize that these mRNAs generally contain a positive-acting uORF that functions to overcome the translational barrier imposed by a second inhibitory uORF located further downstream in response to eIF2α phosphorylation, in the manner established for the positive-acting uORF1 in *GCN4* mRNA, accounting for their increased TE conferred by SM treatment of WT cells. The TE reductions conferred by *eIF2AΔ* exclusively in SM-treated cells would then arise from loss of eIF2A-dependent translation of the stimulatory upstream uORFs in these transcripts required for scanning ribosomes to bypass the downstream inhibitory uORFs when eIF2α is phosphorylated.

To explore this hypothesis further, we focused on the uORF-containing mRNAs that are the most dependent on eIF2A for their translation in SM-treated cells, showing a >1.41-fold reduction in TE in

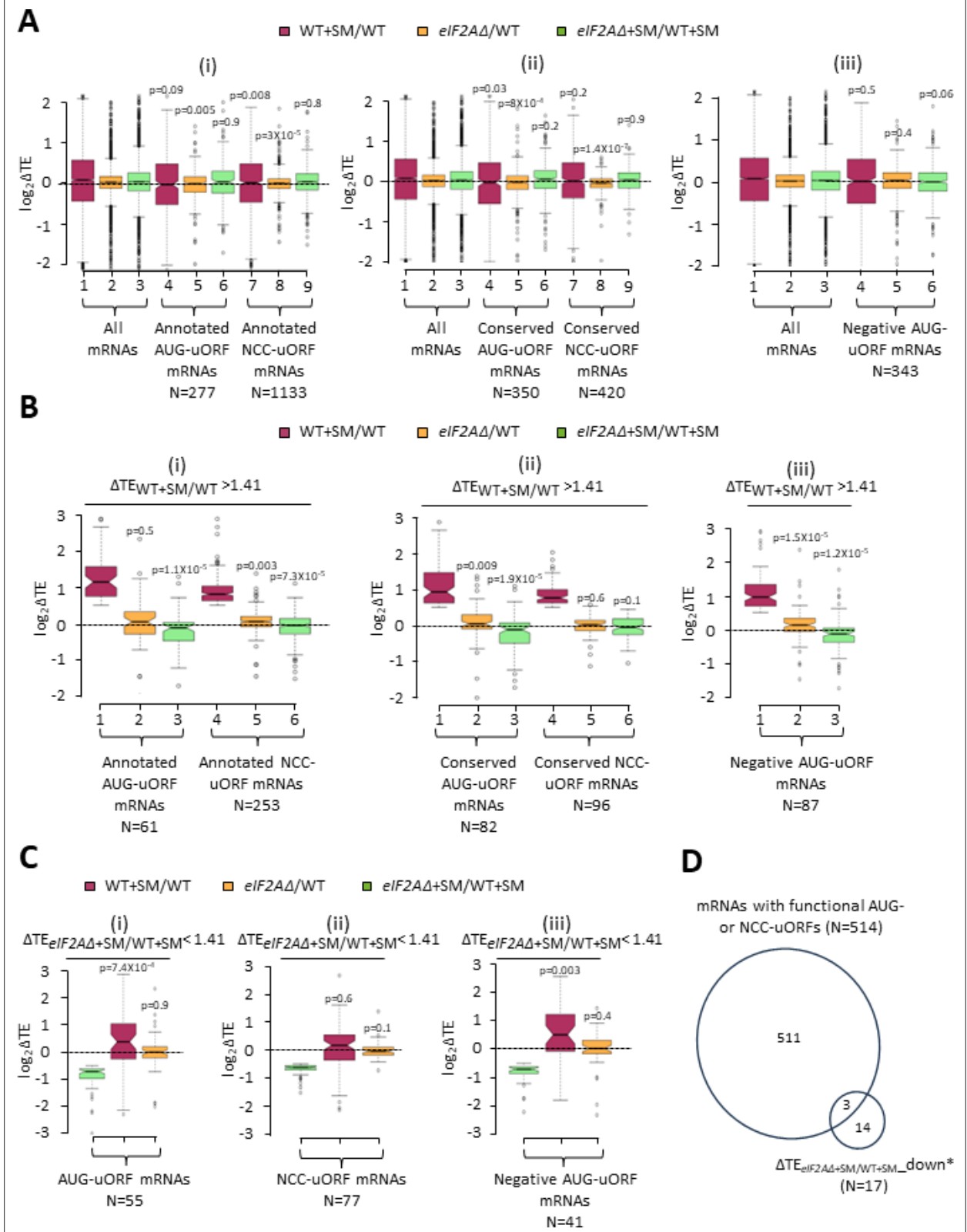

**Figure 6.** Minimal effects of eliminating eIF2A on translation of mRNAs harboring translated upstream open-reading frames (uORFs). (**A**) Notched box plots of $\log_2\Delta TE$ values for all mRNAs (for which translational efficiencies (TEs) could be determined from our ribosome profiling data) containing annotated AUG- or NCC-uORFs (**i**), conserved AUG- or NCC-uORFs (**ii**), or single functional inhibitory AUG-uORFs (**iii**), conferred by sulfometuron methyl (SM) treatment of wild-type (WT) cells (maroon), by the *eIF2AΔ* mutation in untreated cells (orange), or by the *eIF2AΔ* mutation in SM-treated

*Figure 6 continued on next page*

*Figure 6 continued*

cells (green). Statistical significance determined using the Mann–Whitney *U* test is indicated for selective comparisons of changes observed for the indicated groups in comparison to the changes for all mRNAs. A few outliers were omitted from the plots to expand the y-axis scale. (**B**) Notched box plots as in (**A**) for the subsets of the same mRNA groups analyzed there exhibiting >1.41-fold increases in TE in SM-treated versus untreated WT cells. A few outliers were omitted from the plots to expand the y-axis scale. Statistical significance determined as in (**A**). (**C**) Notched box plots as in (**A, B**) for the subsets of the mRNA groups analyzed there exhibiting >1.41-fold decreases in TE in SM-treated *eIF2AΔ* versus SM-treated WT cells. A few outliers were omitted from the plots to expand the y-axis scale. Statistical significance determined as in (**A**). (**D**) Proportional Venn diagram showing overlap between the 17 mRNAs identified in *Figure 3A* showing evidence for a conditional requirement for eIF2A when eIF2 function is reduced by SM (marked in *Figure 3A* with '#'s) and the 514 mRNAs bearing functional AUG or NCC-uORFs.

The online version of this article includes the following source data and figure supplement(s) for figure 6:

**Source data 1.** Spreadsheet 1–3, '*Figure 6A(i)–(iii)*', tabulates the lists and $\log_2\Delta$TE values for the indicated mutant/condition for the annotated AUG- or NCC-uORFs (*Figure 6A(i)*), conserved AUG- or NCC-uORFs (*Figure 6A(ii)*), or single functional inhibitory AUG-uORFs (*Figure 6A(iii)*) (*Figure 6A*).

**Source data 2.** Spreadsheet 1–3, *Figure 6B(i)–(ii)*, tabulates the lists and $\log_2\Delta$TE values as in *Figure 6—source data 1* for the subsets of the same mRNA groups analyzed there exhibiting >1.41-fold increases in translational efficiency (TE) in sulfometuron methyl (SM)-treated versus untreated wild-type (WT) cells (*Figure 6B*).

**Source data 3.** Spreadsheet 1–3, '*Figure 6C(i)–(iii)*', tabulates the lists and $\log_2\Delta$TE values as in *Figure 6—source data 1* for the subsets of the same mRNA groups analyzed there exhibiting >1.41-fold decreases in translational efficiency (TE) in sulfometuron methyl (SM)-treated *eIF2AΔ* versus SM-treated wild-type (WT) cells (*Figure 6C*).

**Source data 4.** Spreadsheet tabulates lists of the 514 mRNAs bearing functional AUG or NCC-uORFs, 17 mRNAs identified in *Figure 3A* showing evidence for a conditional requirement for eIF2A when eIF2 function is reduced by sulfometuron methyl (SM), that is, $\Delta TE_{eIF2A\Delta+SM/WT+SM\_down}$* (N = 17) group, and overlap of mRNAs with functional AUG- or NCC-uORFs (N = 514) and $\Delta TE_{eIF2A\Delta+SM/WT+SM\_down}$ (N = 17) (*Figure 6D*).

**Source data 5.** Spreadsheet tabulates the list of annotated AUG- or NCC-uORFs, chromosome coordinates, start codon of uORF, distances of the uORF AUG from the 5′ end of the mRNA and the main coding sequences (CDS) start codon, and the gene name.

**Source data 6.** Spreadsheet tabulates the list of conserved AUG- or NCC-uORFs, chromosome coordinates, start codon of uORF, distances of the uORF AUG from the 5′ end of the mRNA and the main coding sequences (CDS) start codon, and the gene name.

**Source data 7.** Spreadsheet tabulates the list of the functional uORFs, chromosome coordinates, start codon of uORF, distances of the uORF AUG from the 5′ end of the mRNA and the main coding sequences (CDS) start codon, and the gene name (*May et al., 2023*).

**Figure supplement 1.** eIF2A plays a minimal in regulating upstream open-reading frame (uORF)-mediated translation.

**Figure supplement 1—source data 1.** Spreadsheet 1 tabulates the $\log_2$ ratios of the following parameters for all the expressed annotated uAUG or NCC-uORFs listed in column A in *eIF2AΔ* versus wild-type (WT) cells: relative ribosome occupancy (RRO) in *eIF2AΔ* versus WT cells (RRO Change), RRO for WT, and RRO of *eIF2AΔ* (*Figure 6—figure supplement 1A and E*).

**Figure supplement 1—source data 2.** Spreadsheet 1 tabulates the $\log_2$ ratios of the following parameters for all the evolutionarily conserved expressed uAUG or NCC-uORFs listed in column A in *eIF2AΔ* versus wild-type (WT) cells: relative ribosome occupancy (RRO) in *eIF2AΔ* versus WT cells (RRO Change), RRO for WT, and RRO of *eIF2AΔ* (*Figure 6—figure supplement 1B and F*).

**Figure supplement 1—source data 3.** Spreadsheet 1 tabulates the $\log_2$ ratios of the following parameters for all the expressed annotated uAUG or NCC-uORFs listed in column A in sulfometuron methyl (SM)-treated wild-type (WT) and SM-treated *eIF2AΔ* mutant: relative ribosome occupancy (RRO) in *eIF2AΔ* treated with SM versus WT cells treated with SM (RRO Change), RRO for WT treated with SM, and RRO of *eIF2AΔ* treated with SM (*Figure 6—figure supplement 1C and G*).

**Figure supplement 1—source data 4.** Spreadsheet 1 tabulates the $\log_2$ ratios of the following parameters for all the evolutionarily conserved expressed uAUG or NCC-uORFs listed in column A in *eIF2AΔ* treated with sulfometuron methyl (SM) versus wild-type (WT) cells treated with SM: relative ribosome occupancy (RRO) in *eIF2AΔ* cells treated with SM versus WT cells treated with SM (RRO Change), RRO for WT treated with SM, and RRO of *eIF2AΔ* treated with SM (*Figure 6—figure supplement 1D and H*).

**Figure supplement 1—source data 5.** Spreadsheet 1 tabulates the β-galactosidase activities in units of nmol of ONPG cleaved per mg of protein per min in wild-type (WT) and *eIF2AΔ* strains.

**Figure supplement 2.** Lack of genetic interaction between eIF2A and the purine salvage pathway.

SM-treated *eIF2AΔ* versus SM-treated WT cells. Examining all such mRNAs containing either (i) annotated or conserved AUG-uORFs (*Figure 6C(i)*), (ii) annotated or conserved NCC-uORFs (*Figure 6(ii)*), or (iii) functional inhibitory AUG-uORFs (*Figure 6C(iii)*) revealed that all three groups show TE reductions in response to the *eIF2AΔ* mutation only in the presence of SM (green versus orange data) and, except for the NCC-uORF mRNAs, also show TE increases in response to SM in WT cells (*Figure 6C(i)–(iii)*, maroon data), generally supporting the aforementioned hypothesis. This is the same pattern we observed for all of the uORF-containing mRNAs showing appreciable TE increases on SM treatment of WT cells just described (*Figure 6B(i)–(iii)*). Moreover, this pattern of TE changes was observed above for the group of 32 mRNAs showing highly significant TE reductions conferred by *eIF2AΔ* in

SM-treated cells versus SM treatment of WT (*Figure 3B*); although, as discussed above, only 17 of those transcripts conform to this pattern of TE changes (blue in column 4 but nearly white in column 3 of *Figure 3B*; dubbed $\Delta TE_{eIF2A\Delta+SM/WT+SM\_down*}$). Of these 17 mRNAs, only 3 (*YFL027C, YKL082C,* and *YLR330W/CHS5*) contain a functional AUG- or NCC-initiated uORF (*Figure 6D*), which does not represent a statistically significant enrichment for such mRNAs (p = 0.21). Moreover, our *LUC* reporter and polysome profiling analyses of one of these three transcripts, *YLR330W/CHS5*, failed to confirm a conditional dependence for eIF2A (*Figure 3D and E*). The one mRNA for which conditional eIF2A dependence was confirmed, *YDR420W/HKR1,* does not contain a functional or conserved uORF nor even one merely showing evidence of translation in ribosome profiling experiments. Thus, we could not find compelling evidence that conditional dependence on eIF2A for efficient translation in SM-treated cells is conferred by positive-acting regulatory uORFs.

As an orthogonal approach to detecting a role for eIF2A in regulating the translation of negative-acting uORFs, we reasoned that decreased translation of inhibitory uORFs in *eIF2AΔ* cells would increase the translation of downstream CDSs, leading to a decrease in the ratio of RPFs in uORFs relative to RPFs in the CDSs, which we termed relative ribosome occupancy (RRO). To examine this possibility, we employed DESeq2 to identify statistically significant changes in RRO values for the same groups of mRNAs analyzed above in *Figure 6A and B* containing annotated or conserved uORFs with either AUG- or NCC-start sites, and which also showed evidence of translation in both WT and *eIF2AΔ* cells. We found that no mRNAs exhibited significant changes in RRO in the *eIF2AΔ* mutant versus WT in the absence or presence SM, even at a relatively non-stringent FDR of 0.5 (*Figure 6—figure supplement 1A–D*). Furthermore, the groups of mRNAs with annotated or conserved AUG- or NCC-initiated uORFs showed little or no difference in median RRO in the *eIF2AΔ* mutant versus WT cells in the absence or presence of SM (*Figure 6—figure supplement 1E–H*, yellow versus blue data). At least for the annotated uORFs, the median RRO values are generally higher for mRNAs containing AUG-initiated versus NCC-initiated uORFs in the presence or absence of SM (*Figure 6—figure supplement 1E and G*, blue data), as would be expected from more efficient initiation at AUG

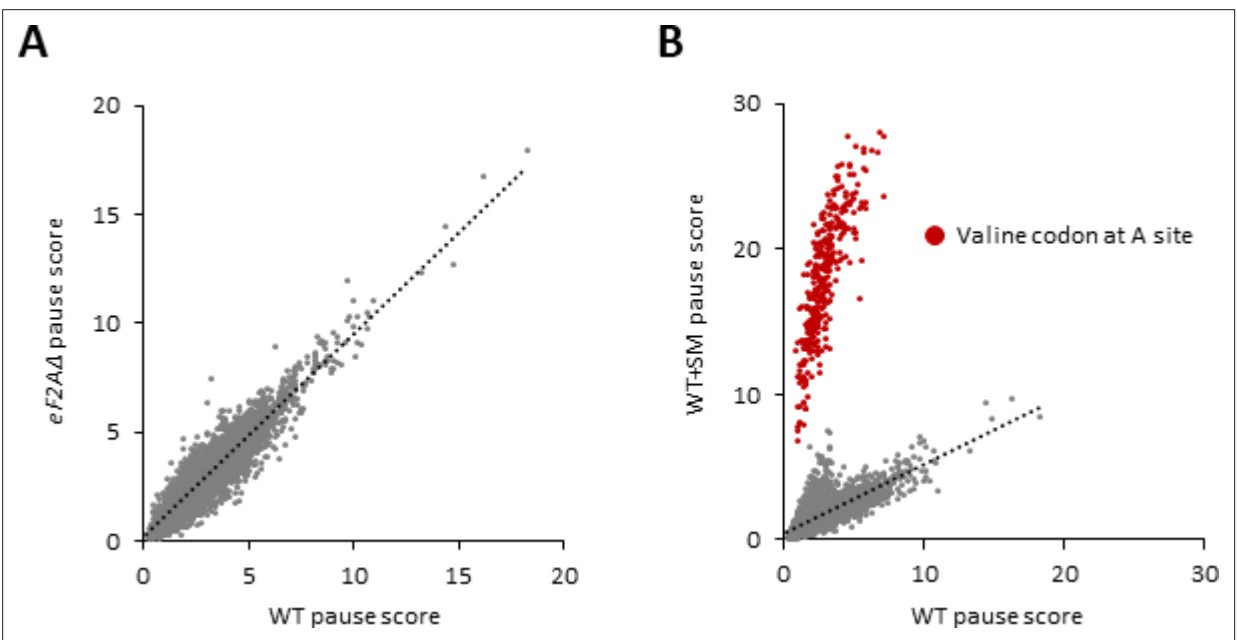

**Figure 7.** eIF2A plays no major role in stimulating translation elongation for particular tripeptide motifs. (**A**) Scatterplot of average pause scores for 8006 tripeptide motifs, comparing the two biological replicates of ribosome profiling data for *eIF2AΔ* versus wild-type (WT) cells. Each dot on the plot represents a tripeptide motif. Pause scores were computed using a shift value of 18 nt from the 3'-end of the footprint, positioning the first codon of the tripeptide motif in the E site. (**B**) Scatterplot of average pause scores for 6267 tripeptide motifs, comparing the two biological replicates of sulfometuron methyl (SM)-treated versus untreated WT cells. All 351 detected motifs with valine codons in the A site are highlighted in red. Pause scores were computed as in (**A**).

The online version of this article includes the following source data for figure 7:

**Source data 1.** Spreadsheet 1 tabulates the pause scores for tripeptide motifs in *eIF2AΔ* cell, wild-type (WT) cells, and SM-treated WT cells.

versus NCC start codons. Thus, our results provide no compelling evidence that eIF2A frequently enhances initiation at negative-acting uORFs initiated by either AUG or NCC start codons in either starved or non-starved cells.

## eIF2A does not affect decoding rates at particular codon combinations during translation elongation

Translation elongation factor eIF5A acts broadly to stimulate the rates of decoding and termination, but is particularly important for certain combinations of codons, which include stretches of proline codons and various three-codon combinations of Proline (Pro), aspartic acid (Asp), and glyine (Gly) codons (*Saini et al., 2009*; *Gutierrez et al., 2013*; *Schuller et al., 2017*). Slower decoding rates for these codon combinations were detected by ribosome profiling of yeast cells depleted of eIF5A by computing pause scores for all ~8000 three-amino acid motifs (*Schuller et al., 2017*). The pause score for each motif is calculated from its 80S occupancy relative to the surrounding stretch of CDS and averaged across the translatome. Analyzing our profiling data revealed no significant increase in pause scores for any tripeptide motif in the *eIF2AΔ* mutant versus WT cells (*Figure 7A*). As a positive control for the analysis, we examined our data on WT cells treated or untreated with SM, observing a marked increase in pause scores for all tripeptide motifs with valine (Val) in the third position, placing the corresponding Val codons in the A-site of the decoding ribosomes (*Figure 7B*). Increased pausing at these motifs is expected from reduced levels of charged valyl-tRNA on inhibition of valine biosynthesis by SM and attendant slower decoding of all Val codons. It is unclear why increased pausing is not found at codons for isoleucine as SM inhibits an enzyme common to the Ile and Val biosynthetic pathways (*Jia et al., 2000*); however, we note that flux through these pathways is differentially regulated and that the Val and leucine pathways compete for a common precursor (*Jones and Fink, 1982*). Taken together, our findings indicate that eIF2A plays no major role in stimulating translation elongation through particular tripeptide motifs.

## Exploring a possible role for eIF2A in purine biosynthesis

As noted above, a synthetic growth phenotype was observed on combining the *eIF2AΔ* mutation with either a temperature-sensitive mutation affecting eIF4E (*Komar et al., 2005*) or deletion of the *FUN12* gene encoding eIF5B (*Zoll et al., 2002*), suggesting functional interactions between eIF2A and these initiation factors. Other genetic interactions involving the *eIF2AΔ* mutation were uncovered in a global analysis of synthetic phenotypes produced by phenotyping double mutants that combine deletions or point mutations in ~90% of all yeast genes (*Costanzo et al., 2016*; *van Leeuwen et al., 2016*), compiled at https://thecellmap.org/costanzo2016/. The profile of genetic interactions observed for the *eIF2AΔ* mutation was found to significantly resemble that of an *ade1Δ* mutation, including synthetic growth defects in *eIF2AΔ* double mutants containing deletions of genes *HPT1* and *FCY2*, whose products function in purine base and cytosine uptake (Fcy2) or in synthesizing purine nucleotides from purine bases via the salvage pathway (Hpt1) (*Ljungdahl and Daignan-Fornier, 2012*). These findings suggested that eIF2A might function as a positive regulator of de novo purine biosynthesis, such that the *eIF2AΔ* mutation would mimic *adeΔ* mutations in reducing growth in combination with the *hpt1Δ* and *fcy2Δ* mutations as a result of simultaneously impairing both the de novo and salvage pathways for adenine biosynthesis (*Ljungdahl and Daignan-Fornier, 2012*). To test this possibility, we generated the *fcy2ΔeIF2AΔ* double mutant and tested it for growth on synthetic minimal (SD) or synthetic complete (SC) medium supplemented with varying concentrations of adenine. Contrary to expectations, we failed to detect any reduction in growth in the double mutant versus either single mutant or the WT strain in the presence or absence of adenine supplements (*Figure 6—figure supplement 2A and B*). We also interrogated our ribosome profiling data to examine the effects of the *eIF2AΔ* mutation on expression of the 17 yeast genes encoding proteins involved in de novo synthesis of inosine, adenine, or guanine nucleotides (*Ljungdahl and Daignan-Fornier, 2012*). The *eIF2AΔ* mutation conferred no significant reductions in the RPFs, mRNA levels, or TEs of any of 17 genes in non-starvation conditions. Moreover, assaying *lacZ* reporters for two such genes, *IMD2* and *IMD3*, in cells cultured under the conditions of our ribosome profiling experiments revealed no significant reductions in expression conferred by elimination of eIF2A (*Figure 6—figure supplement 2C*). Our findings do not support the possibility that eIF2A promotes expression of the biosynthetic genes for de novo purine biosynthesis.

## Discussion

The mammalian eIF2A protein has been implicated in substituting for eIF2 in recruiting Met-tRNAi to certain viral mRNAs that do not require ribosomal scanning for AUG selection under conditions where eIF2α is being phosphorylated and TC assembly is impaired in virus-infected cells. Other studies have identified a role for eIF2A in initiation at near-cognate initiation codons (*Komar and Merrick, 2020*). As the yeast and mammalian eIF2A proteins show considerable sequence similarity, we looked for evidence that the yeast protein could provide a back-up mechanism for Met-tRNAi recruitment when eIF2α has been phosphorylated by Gcn2 in cells starved for isoleucine and valine using the antimetabolite SM. At an SM concentration adequate to reduce bulk polysome assembly and strongly derepress translation of *GCN4* mRNA in WT cells, neither response was detectably altered by the *eIF2AΔ* mutation. Ribosome profiling revealed a broad reprogramming of translational efficiencies conferred by SM treatment of WT cells, noted previously (*Gaikwad et al., 2021*), wherein efficiently translated mRNAs tend to show increased relative TEs at the expense of poorly translated mRNAs, which also appeared to be essentially intact in cells lacking eIF2A. Interrogating the TE changes of individual mRNAs, we found only three in the entire translatome that were significantly altered in *eIF2AΔ* versus WT cells in the absence of SM, with all three showing higher, not lower, TEs in the mutant. Thus, we failed to identify even a single mRNA among the 5340 transcripts with evidence of translation that is dependent on eIF2A for efficient translation in non-stressed cells containing normal levels of TC assembly. Under conditions of SM treatment, we identified 32 mRNAs showing significantly reduced TEs in *eIF2AΔ* versus WT cells, consistent with the possibility that they require eIF2A for Met-tRNAi recruitment only when TC assembly is diminished by eIF2α phosphorylation. However, only 17 of these transcripts showed a pattern of TE changes fully consistent with a conditional requirement for eIF2A under conditions of reduced eIF2 function, exhibiting greater TE decreases when both eIF2 function was impaired by phosphorylation and eIF2A was eliminated from cells. Moreover, we could validate this conditional eIF2A dependence by *LUC* reporter analysis and by measuring TE changes of native mRNAs by polysome profiling for only a single mRNA, *HKR1*. A possible limitation of our *LUC* reporter analysis in *Figure 3D* was the lack of 3′ UTR sequences of the cognate genes, which might be required to observe eIF2A dependence. Given that native mRNAs were examined in the orthogonal assay of polysome profiling in *Figure 3E*, the positive results obtained there for *SAG1* and *SVL3* in addition to *HKR1* should be given greater weight. Nevertheless, our findings indicate a very limited role of yeast eIF2A in providing a back-up mechanism for Met-tRNAi recruitment when eIF2 function is diminished by phosphorylation of its α-subunit.

We also looked intensively at groups of mRNAs containing uORFs to determine if eIF2A might promote uORF translation and thereby influence translation of the downstream CDSs. As observed for *GCN4* mRNA, eliminating eIF2A had no effect on the TEs of *CPA1* and *CAD1* mRNAs, which contain single inhibitory uORFs that appear to be suppressed by eIF2α phosphorylation. We interrogated three different compilations of yeast mRNAs containing translated uORFs containing AUG or NCC start codons, including those with functional evidence of controlling translation of the downstream CDSs or showing evolutionary sequence conservation and, once again, found little evidence for a significant role of eIF2A in either increasing or decreasing the TEs of such transcripts. It remains to be seen if there are any uORFs in yeast mRNAs that depend on eIF2A for their translation. We also found no evidence indicating that eIF2A inhibits utilization of IRES elements identified previously in *URE2*, *PAB1*, and *GIC1* mRNAs; although, our results could be explained if the IRESs make only small contributions to the overall translation of these mRNAs. Finally, we found no evidence that eIF2A affects the rate of decoding of any particular tripeptide motifs in the elongation phase of protein synthesis.

The lack of evidence in our study for eIF2A function in translation initiation stands in contrast to findings of negative genetic interactions between the *eIF2AΔ* and mutations in yeast eIF4E (*Komar et al., 2005*) and eIF5B (*Zoll et al., 2002*) wherein combining the mutations conferred a more severe cell growth defect. Synthetic negative interactions between *eIF2AΔ* and deletions of each of the two genes encoding eIF4A (*TIF1* and *TIF2*) were also reported in a global analysis of synthetic phenotypes observed in double mutants that combine deletions or point mutations in ~90% of all yeast genes (*Costanzo et al., 2016*; *van Leeuwen et al., 2016*), compiled at https://thecellmap.org/costanzo2016/. One possibility is that eIF2A is functionally redundant with these initiation factors rather than with eIF2. Another is that eliminating eIF2A affects a different cellular process and that

expression of one or more genes involved in that process is diminished by mutations in eIF4E, –4A, and –5B.

A recent study indicated that human recombinant eIF2A inhibits translation in rabbit reticulocyte lysate (RRL) of all mRNAs tested, including one driven by the cricket paralysis virus IGR IRES that requires no canonical initiation factors (*Grove et al., 2023*). Because the protein binds to 40S subunits and the addition of excess 40S subunits to RRL mitigates its inhibitory effect, it was concluded that human eIF2A sequestered 40S subunits in an inactive complex. While this finding is consistent with the conclusion that yeast eIF2A represses IRES-driven translation in yeast, the finding that *eIF2AΔ* does not increase bulk translation or impact yeast cell growth obtained here and elsewhere (*Zoll et al., 2002*) suggests that eIF2A does not appreciably sequester 40S ribosomes in yeast cells, even in stress conditions of eIF2α phosphorylation (*Figure 1A*). The similar finding that knocking out the eIF2A gene had no detectable effect on bulk translation in keratinocytes (*Sendoel et al., 2017*) led *Grove et al., 2023* to surmise that mammalian eIF2A is not normally present in the cytoplasm and would have to be released from a different cellular compartment to impact translation (*Grove et al., 2023*).

The profile of genetic interactions observed for the *eIF2AΔ* mutation in the aforementioned global analysis of synthetic phenotypes was found to resemble that of an *ade1Δ* mutant (*Costanzo et al., 2016*; *van Leeuwen et al., 2016*), lacking an enzyme of de novo biosynthesis of AMP and GMP, which suggested that eIF2A might be a positive effector of purine biosynthesis. However, we could not confirm the reported negative genetic interaction between the *eIF2AΔ* and *fcy2Δ* mutations, and our ribosome profiling data revealed no effect of *eIF2AΔ* on the RPF or mRNA abundance of the 17 genes involved in de novo purine biosynthesis.

It is surprising that we found so little evidence that eIF2A functions in the yeast translatome even when the activity of the canonical factor eIF2 is attenuated by amino acid starvation. It is possible that eIF2A plays a more important role in translation in some other stress condition, for example, one in which it is released into the cytoplasm (*Grove et al., 2023*) or when eIF4E or eIF5B function is attenuated, or that it functions in a different cellular process altogether in budding yeast. eIF2A may have acquired a new role in controlling translation initiation during mammalian evolution, or it could have lost its function in translation and acquired a different one during fungal evolution.

## Materials and methods
### Yeast strain and plasmid construction
Yeast strains used in this study are listed in *Table 1*. The primers used for strain construction and verification are listed in Table 3. Strain SGY3 (*eIF2AΔfcy2Δ*) was generated through a two-step process. First, the *kanMX* cassette of the *eIF2AΔ* deletion allele, *ygr054wΔ::kanMX4,* in strain F2247 was swapped with a hygromycin-resistance cassette to produce SGY1 (*ygr054wΔ::hphMX4*) by transforming F2247 with a DNA fragment containing the *hphMX4* allele that was PCR-amplified from plasmid p4430. Transformants were selected on YPD agar plates supplemented with 300 µg/mL hygromycin B. In the second step, the *FCY2* gene was deleted in SGY1 by transformation with a DNA fragment containing the *fcy2Δ::kanMX4* allele PCR-amplified from the genomic DNA of strain F2379. Transformants were selected on YPD agar plates containing 200 µg/mL geneticin (G418) to produce SGY3. The *ygr054wΔ::hphMX4* and *fcy2Δ::kanMX4* alleles in strains SGY1 and SGY3, respectively, were verified by PCR analysis of chromosomal DNA using the appropriate primers listed in Table 3.

**Table 1.** Yeast strains used in this study.

| Strain | Genotype | Source or reference |
| --- | --- | --- |
| BY4741 | *MATa his3Δ1 leu2Δ0 met15Δ0 ura3Δ0* | GE Healthcare |
| BY4742 | *MATα his3Δ1 leu2Δ0 lysΔ2 ura3Δ0* | GE Healthcare |
| F2247/4684 | *MATa his3Δ1 leu2Δ0 met15Δ0 ura3Δ0 ygr054wΔ::kanMX4* | GE Healthcare |
| F2379/191 | *MATa his3Δ1 leu2Δ0 met15Δ0 ura3Δ0 fcy2Δ::kanMX4* | GE Healthcare |
| YSG1 | *MATa his3Δ1 leu2Δ0 met15Δ0 ura3Δ0 ygr054wΔ::hphMX4* | This study |
| YSG3 | *MATa his3Δ1 leu2Δ0 met15Δ0 ura3Δ0 fcy2Δ::kanMX4 ygr054wΔ::hphMX4* | This study |

**Table 2.** Plasmids used in this study.

| Name | Description | Source/reference |
|---|---|---|
| p6586 | IMD2-lacZ | *Escobar-Henriques and Daignan-Fornier, 2001* |
| p6587 | IMD3-lacZ | *Escobar-Henriques and Daignan-Fornier, 2001* |
| p4430 | hphMX4 | *Goldstein and McCusker, 1999* |

*F.LUC* reporter plasmids containing native promoter and 5' UTR (nucleotides (nt) indicated) and first 20 codons of the main CDSs of the indicated genes

| Name | Description | Source/reference |
|---|---|---|
| p6593 | 600 nt 5' ncDNA + first 20 codons of *SVL3* in pRS416 | This study |
| p6594 | 600 nt 5' nc DNA + first 20 codons of *NET1* in pRS416 | This study |
| p6595 | 1080 nt 5' ncDNA + first 20 codons of *RAD9* in pRS416 | This study |
| p6596 | 1080 nt 5' ncDNA + first 20 codons of *SAG1* in pRS416 | This study |
| p6597 | 300 nt 5' ncDNA + first 20 codons of *PUS7* in pRS416 | This study |
| p6598 | 720 nt 5' ncDNA + first 20 codons of *DBF4* in pRS416 | This study |
| p6599 | 420 nt 5' ncDNA + first 20 codons of *PBP4* in pRS416 | This study |
| p6600 | 600 nt 5' ncDNA + first 20 codons of *HKR1* in pRS416 | This study |
| p6601 | 840 nt 5' ncDNA + first 20 codons of *MET4* in pRS416 | This study |
| p6602 | 360 nt 5' ncDNA + first 20 codons of *CHS5* in pRS416 | This study |
| p6603 | 540 nt 5' ncDNA + first 20 codons of *DNA2* in pRS416 | This study |
| p6604 | 420 nt 5' ncDNA + first 20 codons of *NSI1* in pRS416 | This study |

All plasmids employed in this study are listed in *Tables 2* and *3*. Plasmids p6593-p6604 were constructed by LifeSct LLC by synthesizing DNA fragments containing the promoter, 5′ UTR, and first 20 codons of the main CDS of the candidate genes and using them to replace the corresponding fragment of p6029 (*Sen et al., 2015*), fusing the first 20 codons of the candidate genes to the *F.LUC* CDS.

## Preparation of ribosome footprint (RPF) and RNA-seq sequencing libraries

Ribosome profiling and RNA-seq analyses were carried out in parallel on strains BY4742 (WT) and F2247 (*eIF2AΔ*), with two biological replicates for each strain, as described previously (*Gaikwad et al., 2021*). In brief, both strains were grown at 30°C in SC until reaching log phase at $A_{600}$ = 0.5–0.6 for untreated cells, or to $A_{600}$ = 0.5–0.6 in SC lacking isoleucine and valine (SC-Ile/Val) and treated with SM at 1 µg/mL for 25 min for SM-treated cells. Cells were harvested by high-speed vacuum filtration and snap-frozen in liquid nitrogen. Cell lysis was performed using a freezer mill in the presence of lysis buffer containing 500 µg/mL of cyclohexamide, and RPFs were prepared by digesting cell lysates with RNase I. The 80S monosomes were resolved by sedimentation through a 10–50% sucrose gradient. The RPFs were purified from the monosomes using hot phenol-chloroform extraction. After size selection and dephosphorylation steps, a Universal miRNA cloning linker was attached to the 3' ends of the footprints. This was followed by reverse transcription, circular ligation, rRNA subtraction, PCR amplification of the cDNA library, and DNA sequencing using an Illumina HiSeq system at the NHLBI DNA Sequencing and Genomics Core at NIH (Bethesda, MD).

For RNA-seq library preparation, total RNA was extracted and purified from aliquots of the same snapped-frozen cells lysates described above using hot phenol-chloroform extraction. Then, 5 µg of randomly fragmented total RNA was used for library generation and sequencing, similar to the steps mentioned above except that the Ribo-Zero Gold rRNA Removal Kit (Illumina; MRZ11124C) was employed to remove rRNA after linker-ligation.

**Table 3.** Primers used in this study.

| Primer name | Primer sequence 5′ – 3′ |
| --- | --- |
| Primers for qRT-PCR | |
| Firefly Fwd | GTGTTGGGCGCGTTATTTATC |
| Firefly Rev | TAGGCTGCGAAATGTTCATACT |
| CHS5 Fwd | CTGTGGAGGATGCCAATGAA |
| CHS5 Rev | AGAGGCAATGTCGGTAGTAAAC |
| DBF4 Fwd | GCAAGGCAAGAAACTGAAGAAG |
| DBF4 Rev | GATGTGCACCACTTGCTTTG |
| HRK1 Fwd | GCCATACAGCTCTGTCCATT |
| HRK1 Rev | AGAGGAAACGGATGCAGATAAG |
| NET1 Fwd | TGCTTCAGCTTCTTCCTCTTC |
| NET1 Rev | GGTTACGGGTGGTTTCCTATT |
| SAG1 Fwd | CCATCCAGTCCCTCATCTTATAC |
| SAG1 Rev | TGGCACAGAAGGCGTAAA |
| SVL3 Fwd | CCTCCTACAACCTCTGTTTCAG |
| SVL3 Rev | GGTACTAGGCGAAGCCATATTC |
| 18S rRNA Fwd | TCACCAGGTCCAGACACAATAAG |
| 18S rRNA Rev | TCTCGTTCGTTATCGCAATTAAGC |
| ACT1 Fwd | TGTGTAAAGCCGGTTTTGCC |
| ACT1 Rev | GATACCTCTCTTGGATTGAGCTTC |
| Primers used for deletion verification and construction of double deletion strains | |
| KanB | CTGCAGCGAGGAGCCGTAAT |
| KanC | TGATTTTGATGACGAGCGTAAT |
| Hyg Fwd | CGGATCCCCGGGTTAATTAA |
| HygC Rev | GAATTCGAGCTCGTTTAAAC |
| Hyg-B | TTTCGATCAGAAACTTCTCGACA |
| Hyg-C | TGCTCCGCATTGGTCTTGACC |
| eIF2A-A | TTCAGCTTCATAGCGATTTATTTTC |
| eIF2A-B | CATATGATTAATTGAGCCGGTTTAC |
| eIF2A-C | AGAGTACATAAGTCAACACCCAAGC |
| eIF2A-D | GACACTCCATATTCATTTATTGCCT |
| FCY2-A | TATCATTTCCGCTTATCTGACTTCT |
| FCY2-B | CTAACCTTAACACCAACTTCCTCAA |
| FCY2-C | TTGAGGAAGTTGGTGTTAAGGTTAG |
| FCY2-D | AATCAGCAGATTCCATCAAAAGTAG |

## Differential gene expression and uORF translation analysis of ribosome profiling data

Processing and analysis of sequence libraries of RPFs or total mRNA fragments, including Wiggle track normalization to the total number of mapped reads for viewing RPF or RNA reads in the IGV browser, were conducted exactly as described previously (*Gaikwad et al., 2021*). In brief, sequencing reads

were trimmed and noncoding RNA sequences were eliminated by aligning trimmed FASTA sequences to the *Saccharomyces cerevisiae* ribosomal database using Bowtie (*Langmead et al., 2009*). The remaining reads were mapped to the yeast genome using TopHat (*Trapnell et al., 2009*), and DESeq2 was used for statistical analysis of mRNA reads, RPFs, and TEs (*Love et al., 2014*). The R script employed for DESeq2 analysis of TE changes can be found on GitHub (https://github.com/hzhangh-enry/RiboProR; *Zhang, 2023*). Wiggle files were visualized using IGV 2.4.14 at http://software.broa-dinstitute.org/software/igv/ (*Robinson et al., 2011*).

RROs for genes containing translated uORFs were calculated by dividing the RPF counts in the uORF by the RPF counts of the main CDS. DESeq2 was employed to conduct statistical analysis on changes in RRO values between the two biological replicates of different genotypes. uORFs with mean RPF counts below 2 or mean CDS RPF counts below 32 in the combined samples (two replicates each) were excluded from the analysis. The compilations of annotated translated uORFs (*Martin-Marcos et al., 2017*), evolutionarily conserved translated uORFs (*Spealman et al., 2018*), or functional uORFs whose elimination by start codon mutation conferred a consistent and statistically significant alter-ation in translation of the downstream GFP CDSs in reporter constructs (*May et al., 2023*) were all published previously and are provided in *Figure 6—source data 5–7*.

Tripeptide pause scores were computed as described previously (*Meydan et al., 2023*). Briefly, this involved dividing the 80S reads per million (rpm) of a three-amino acid motif by the average rpm in the surrounding region (± 50 nt around each motif). Sites smaller than the ± 50 nt window were excluded from the analysis. To generate average pause scores, we calculated the mean of individual pause scores for each tripeptide motif across the translatome. Motifs represented in the genome less than 100 times were excluded to reduce noise in the analysis.

## Yeast biochemical methods

β-Galactosidase activities were assayed in WCEs using a modified version of a protocol described previously (*Moehle and Hinnebusch, 1991*) in which the cleavage of ONPG was determined manu-ally after a 30 min incubation with the WCE. Expression of luciferase was assayed in WCEs as previ-ously described (*Gaikwad et al., 2021*) using the Dual-Luciferase Reporter Assay System (Promega) following the supplier's protocol, and luciferase activities were normalized to total protein levels in the WCEs determined using the Bradford assay kit (Bio-Rad Laboratories).

## Polysome profile analysis and measurement of TEs of individual mRNAs

To conduct bulk polysome profiling, cells were cultured as described in the figure legends, harvested by centrifugation, and WCEs were prepared by vortexing cell pellets, applying eight cycles of vortexing for 30 s followed by incubation on ice for 30 s, using two volumes of glass beads in ice-cold 1× breaking buffer (20 mM Tris–HCl (pH 7.5), 50 mM KCl, 10 mM MgCl$_2$, 1 mM DTT, 200 µg/mL heparin, 50 µg/mL cycloheximide, and 1 cOmplete EDTA-free Protease Inhibitor cocktail Tablet (Roche)/50 mL buffer). Fifteen A$_{260}$ units of WCEs were layered on a pre-chilled 10–50% sucrose gradient and centri-fuged at 40,000 rpm for 2 hr at 4°C in a SW41Ti rotor (Beckman). Gradient fractions were continuously scanned at A$_{260}$ using the BioComp Gradient Station, and polysome to monosome (P/M) ratios were calculated using ImageJ software for at least two biological replicates.

To determine polysome association of individual mRNAs and calculate their TEs, 20 A$_{260}$ units of WCEs from each of three replicate cultures were sedimented through a 10–50% sucrose gradient by centrifugation at 35,000 rpm for 2.4 hr and the gradient fractions were scanned at A$_{260}$ nm. The frac-tions containing 80S monosomes or different polysomal species (2-mers, 3-mers, etc.) were collected using 'the advanced fraction' function of the BioComp fractionator (*Figure 3—figure supplement 1*). RNA was extracted from 1/5th of the total volume of input WCEs and from 300 µL of each set of pooled fractions containing 80S or different polysomal species using QIAzol lysis Reagent (QIAGEN) according to the manufacturer's protocol. Reverse transcription (RT) was done using SuperScript III First-Strand Synthesis SuperMix (Invitrogen) with random hexamers, using 5 µg of RNA for each reverse transcription (RT) reaction. Transcript levels were quantified by qPCR using the Brilliant III Ultra-Fast SYBR Green qPCR Master Mix (Agilent Technologies) on a Roche LightCycler 96 Instrument. Primers listed in *Table 3* were used to measure the individual mRNAs and 18S rRNA. The absolute mRNA/18S rRNA levels were calculated using the $2^{-CT}$ method and corrected to account for differ-ences in the proportions of the pooled fractions from which RNA was isolated by multiplying by a

factor calculated by dividing the total fraction volume (in μL) by 300 μL. They were further corrected to account for differences in the proportion of the total RNA employed for RT by multiplying by a factor calculated as 25 μL (the total volume of extracted mRNA) divided by the volume of the pooled fraction (in μL) containing 5 μg of RNA. To account for potential losses during RNA extraction, RNA recovery normalization factors were calculated for each fraction by determining the proportion of total $A_{260}$ units across all gradient fractions that is present in each set of pooled fraction (calculated from the $A_{260}$ trace obtained during gradient fractionation) and dividing the results by the proportion of total 18S rRNA across the gradient found in the corresponding pooled fraction. A second normalization factor was calculated to correct for losses in recovery of monosomes/polysomes in the gradient separations of fixed amounts of input WCEs by calculating the total $A_{260}$ units found in monosomes/polysomes for each gradient, determining the mean value for all of the gradients/samples analyzed in parallel, and dividing the mean value by the value determined for each gradient. (This correction assumes that WT and *eIF2AΔ* cells have the same total amounts of monosomes/polysomes per $A_{260}$ of WCE, as indicated by our repeated polysome profiling of biological replicates depicted in *Figure 1A*). The absolute amounts of mRNAs measured in each pooled fraction were multiplied by both normalization factors to obtain the normalized mRNA levels for each pooled fraction, which was multiplied by the number of ribosomes per mRNA in that pooled fraction (i.e., one for monosomes, two for 2-mer polysomes, etc.) and summed across the gradient fractions to calculate the total number of ribosomes translating the mRNA. To calculate TEs, this last quantity was divided by amount of input mRNA measured in the starting WCE normalized to level of *ACT1* mRNA. For each of three biological replicate cultures of each strain, WT or *eIF2AΔ* mutant, we determined the ratio of TE in the presence versus absence of SM and calculated the mean and SEM values for ΔTEs.

## Statistical analyses and data visualization

Notched box plots were created using a web-based tool (http://shiny.chemgrid.org/boxplotr/). Scatterplots and volcano plots were generated using the scatterplot function in Microsoft Excel. Hierarchical cluster analysis of TE changes in mutants/conditions was performed using the R heatmap.2 function from the 'gplots' library using the default hclust hierarchical clustering algorithm. Smoothed scatterplots were computed and plotted using the ggplot2 package in R. Calculation of Spearman's correlation coefficients and Student's *t*-tests were performed using built-in features of Microsoft Excel. The Mann–Whitney *U* test and p-values for Pearson's correlation were computed using the R Stats package in R.

## Acknowledgements

We thank Sezen Meydan for performing the tripeptide pause score analysis and Bertrand Daignan-Fornier for gifts of reporter plasmids. We are grateful to the members of our laboratory and those of the Lorsch, Dever, and Guydosh labs for many helpful suggestions. This work was supported by the Intramural Research Program of the National Institutes of Health.

## Additional information

### Competing interests

Alan G Hinnebusch: Reviewing editor, *eLife*. The other authors declare that no competing interests exist.

### Funding

| Funder | Grant reference number | Author |
|---|---|---|
| Intramural Research Program, Eunice Kennedy Shriver National Institute of Child Health and Human Development | HD001004-32 | Fardin Ghobakhlou |

| Funder | Grant reference number | Author |
|--------|------------------------|--------|

The funders had no role in study design, data collection and interpretation, or the decision to submit the work for publication.

## Author contributions

Swati Gaikwad, Conceptualization, Data curation, Formal analysis, Validation, Investigation, Visualization, Writing - original draft; Fardin Ghobakhlou, Investigation; Hongen Zhang, Software, Formal analysis; Alan G Hinnebusch, Conceptualization, Formal analysis, Supervision, Funding acquisition, Project administration, Writing - review and editing

## Author ORCIDs

Swati Gaikwad (i) http://orcid.org/0000-0002-1438-9497
Fardin Ghobakhlou (i) https://orcid.org/0000-0002-1391-2012
Hongen Zhang (i) http://orcid.org/0000-0001-6871-8463
Alan G Hinnebusch (i) http://orcid.org/0000-0002-1627-8395

Reviewer #1 (Public Review): https://doi.org/10.7554/eLife.92916.3.sa1
Reviewer #2 (Public Review): https://doi.org/10.7554/eLife.92916.3.sa2
Reviewer #3 (Public Review): https://doi.org/10.7554/eLife.92916.3.sa3
Author Response https://doi.org/10.7554/eLife.92916.3.sa4

---

# Additional files

## Supplementary files

• MDAR checklist

## Data availability

Sequencing data from this study have been deposited to the NCBI Gene Expression Omnibus (GEO) under GEO accession number GSE241473. All other data generated or analysed during this study are included in the manuscript and supporting files; Source Data files have been provided for *Figures 2, 3, 6 and 7*, *Figure 2—figure supplement 1*, *Figure 6—figure supplement 1*.

The following dataset was generated:

| Author(s) | Year | Dataset title | Dataset URL | Database and Identifier |
|-----------|------|---------------|-------------|-------------------------|
| Gaikwad S, Ghobakhlou F, Zhang H, Hinnebusch AG | 2024 | Yeast eIF2A plays a minimal role in translation initiation in vivo | https://www.ncbi.nlm.nih.gov/geo/query/acc.cgi?acc=GSE241473 | NCBI Gene Expression Omnibus, GSE241473 |

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
