## [Editor Report · eLife assessment]

In this **important** study, Gaikwad and colleagues employed ribosome profiling in conjunction with standard biochemical approaches to investigate the role of eIF2A in translation initiation in yeast under optimal growth conditions or stress. The authors provide **convincing** data that eIF2A is not implicated in translation initiation in yeast, a finding that is anticipated to inspire future investigations to identify the cellular role(s) of eIF2A in yeast. Considering the broad scope of cellular functions attributed to eIF2A, this study should be of interest to a wide spectrum of biomedical researchers ranging from those studying mechanisms of translation regulation to virologists and cancer biologists.

---

## [Referee Report · Reviewer #1 (Public Review)]

Summary:

The study follows the role of yeast eIF2A protein as potential translation initiation factor engaged in the non-canonical translation initiation under stress conditions and as a substitute for eIF2. Using ribosome profiling, RNA-Seq and reporter based assays authors evaluated the role of eIF2A protein under regular or stress conditions (cells starved for branched amino acids). Authors found that yeast cells depleted of eIF2A protein do not change significantly their translation initiation, or translation in general. In the contrast to previously reported data for human homolog yeast eIF2A does not significantly contribute to regulation of the uORFs, regardless if they start with canonical AUG or near cognate start codons. eIF2A is not involved in the repression of IRES element in URE2 gene or has a role in purine biosynthesis. It appears that in yeast eIF2A contributes to regulation of very limited number of mRNAs (32 with significant changes in translation efficiency), where only 17 of such messages indeed are consistent with eIF2A deletion and single mRNA (HKR1) could be validated in reporter assay.

Strengths:

The strength of the manuscript is complete analysis and unbiased approach using genomic analysis methods (ribosome profiling and RNA-seq) as well as reporter validation studies. Additional strength of the manuscript is scientific rigor and statistics associated with data analyses, clear data presentation and discussion of the results in the context of the previous studies and results.

Weaknesses:

none noted

---

## [Referee Report · Reviewer #2 (Public Review)]

Summary:

Gaikwad et al. investigated the role of eIF2A in translational response to stress in yeast. For this purpose, the authors conducted ribosome profiling under SM treatment in eIF2A-depleted strain. Data analysis revealed that eIF2A did not influence translation from mRNAs bearing uORFs or cellular IRESes, in the stress condition, broadly. The authors found that only a small number of mRNAs were supported by eIF2A. The data should be helpful for researchers in the fields.

Major points:

1. The weakness of this work is the lack of clarification on the function of eIF2A in general. The novelty of this study was limited.

2. Related to this, it would be worth investigating common features in mRNAs selectively regulated (surveyed in Figure 3A). Also, it would be worth analyzing the effect of eIF2A deletion on elongation (ribosome occupancy on each codon and/or global ribosome footprint distribution along CDS) and termination/recycling (footprint reads on stop codon and on 3′ UTR).

3. Regarding Figure 3D, the reporters were designed to include promoter and 5′ UTR of the target genes. Thus, it should be worth noting that reporter design was based on the assumption that eIF2A-dependency in translation regulation was not dependent on 3′ UTR or CDS region. The reason why the effects on ribosome profiling-supported mRNAs could not be recapitulated in reporter assay may originate from this design. This should be also discussed.

4. Related to the point above, the authors claimed that eIF2A affects "possibly only one" (HKR1) mRNA. However, this was due to the reporter assay which is technically variable and could not allow some of the constructs to pass the authors' threshold. Authors may be worth considering better wording for this point.

5. For Figure 3D, it would be worth considering to test all the #-marked genes (in Figure 3C) in this set up.

6. In box plots, the authors should provide the statistical tests, at least where the authors explained in the main text.

---

## [Referee Report · Reviewer #3 (Public Review)]

Summary:

The authors have undertaken a study to rigorously characterize the possible role of eIF2A in regulating translation in yeast. The authors test for a role of eIF2A in the absence or presence of cellular stress and conclude that eIF2A does not play any significant role in regulating translation initiation in yeast.

The authors have used rigorous experimental approaches, including genome wide ribosome profiling analysis in the absence or presence of stress, to show that eIF2A does not function in translation initiation on most mRNAs in yeast. Interestingly, the authors do identify a small number of mRNAs that possess some eIF2A dependency, so they constructed reporters to rigorously test them. One mRNA, HKR1, appears to possess a degree of eIF2A-dependent translation regulation.

No role of eIF2A in translation initiation is apparent and one limitation of the study is that the authors do not determine what function eIF2A plays in yeast.

---

## [Author Response]

The following is the authors’ response to the original reviews.

**Reviewer #1:**

We thank the referee for the positive review.

**Reviewer #2 (Public review):**

We thank the referee for his/her constructive comments

1. The weakness of this work is the lack of clarification on the function of eIF2A in general. The novelty of this study was limited.

We believe our study is valuable in providing strong evidence that eIF2A does not functionally substitute for eIF2 in tRNAi recruitment even when eIF2 function is impaired, and in showing that it does not contribute to translational control by uORFs or IRESs, thus ruling out the most likely possibilities for its function in yeast based on studies of the mammalian factor. We agree that the function of yeast eIF2A remains to be identified; however, we think this should be regarded as a limitation rather than a weakness in experimental design or data obtained in the current study.

1. Related to this, it would be worth investigating common features in mRNAs selectively regulated (surveyed in Figure 3A).

We did not embark on this because only 17 of the 32 transcripts showing TE reductions in Fig. 3A showed a pattern of TE changes consistent with a conditional requirement for eIF2A under conditions of reduced eIF2 function, exhibiting greater TE decreases when both eIF2 function was impaired by phosphorylation and eIF2A was eliminated from cells. Moreover, we could validate this conditional eIF2A dependence by LUC reporter for only a single mRNA, HKR1.

Also, it would be worth analyzing the effect of eIF2A deletion on elongation (ribosome occupancy on each codon and/or global ribosome footprint distribution along CDS) and termination/recycling (footprint reads on stop codon and on 3′ UTR).

We have analyzed the effects of deleting eIF2A on ribosome pausing at individual codons by calculating tri-peptide pause scores from our ribosome profiling data. The results shown in new Fig. 7 reveal that eIF2A plays no discernible role in stimulating the rate of decoding of any three-codon combinations.

1. Regarding Figure 3D, the reporters were designed to include promoter and 5′ UTR of the target genes. Thus, it should be worth noting that reporter design was based on the assumption that eIF2A-dependency in translation regulation was not dependent on 3′ UTR or CDS region. The reason why the effects on ribosome profiling-supported mRNAs could not be recapitulated in reporter assay may originate from this design. This should be also discussed.

We agree and included this stipulation in the DISCUSSION, while at the same time noting that the native mRNAs were examined in the orthogonal assay of polysome distributions.

1. Related to the point above, the authors claimed that eIF2A affects "possibly only one" (HKR1) mRNA. However, this was due to the reporter assay which is technically variable and could not allow some of the constructs to pass the authors' threshold. Alternative wording for this point should be considered.

We agree and revised text in the DISCUSSION to read: “A possible limitation of our LUC reporter analysis in Fig. 3D was the lack of 3’UTR sequences of the cognate transcripts, which might be required to observe eIF2A dependence. Given that native mRNAs were examined in the orthogonal assay of polysome profiling in Fig. 3E, the positive results obtained there for SAG1 and SVL3 in addition to HKR1 should be given greater weight. Nevertheless, our findings indicate a very limited role of yeast eIF2A in providing a back-up mechanism for Met-tRNAi recruitment when eIF2 function is diminished by phosphorylation of its α-subunit.”

1. For Figure 3D, it would be worth considering testing the #-marked genes (in Figure 3C) in this set up.

Actually, we did test 10 of the 17 mRNAs marked with “#”s in the reporter assays of Fig. 3C, which had been noted in the Fig. 3C legend.

1. In box plots, the authors should provide the statistical tests, at least where the authors explained in the main text.

At the first occurrence of a notched box plot (Fig. 2D), we explained in the main text that in all such plots, when the notches of different boxes do not overlap, their median values differ significantly with a 95% confidence level. In cases where overlaps between notches is difficult to assess by eye, we added the results of Mann-Whitney U tests with the p values indicated by asterisks, as explained in the legends. We added results of additional Mann-Whitney U tests to such box plots in Figs. 3B, 6A-C, and 6-supp. 1E & G and mentioned this in the corresponding legends.

**Reviewer #2 (Recommendations For The Authors):**
The first section of "Yeast eIF2A does not play a prominent role as a functional substitute for eIF2 in the presence or absence of amino acid starvation" can be subdivided into a couple of sections for better readability.

Done.

Although the authors have used SM to induce ISR in yeasts previously, the validation of eIF2alpha phosphorylation in Western blot would be helpful for readers. Also, it should be worth testing whether eIF2alpha phosphorylation was properly induced in eIF2A KO cells.

The translational induction of GCN4 mRNA, which we have documented in WT and eIF2A∆ cells, provides a quantitative read-out of eIF2 functional attenuation superior to determining the proportion of eIF2α that is phosphorylated.

For Figure 2B, the Venn diagram that shows the overlap between TE-changes genes in WT_SM/WT and those in eIF2A∆_SM/eIF2A∆ would be helpful (although a list was provided by the source data).

The Venn diagram has been provided in a new figure, Figure 2-figure supplement 1B.

For Figures 1C and 5A-B, the depiction of the positions of uORFs within the orange gene region would be helpful for readers.

Done.

For Figure 4A-C, the depiction of the IRES regions (if known) within the orange gene region would be helpful for readers.

Done for the URE2 IRES, whose location is known.

For Figures 1C, 4A-C, and 5A-B, the y-axis should have a label/scale.

Added.

For Figure 3C, the definition of #-marked genes should be concretely described (e.g., value range) in the legend.

Added.

For Figure 3D-E, the statistical test has been only shown in a couple of data. A full depiction of the statistical results for all the data sets may be helpful for readers.

We explained that when notches in box plots do not overlap, their medians differ with 95% confidence. In cases where overlaps were difficult to discern, we added p values from Mann-Whitney U tests to the relevant box plots.

For Figure 3E, it would be helpful if the authors could show the UV spectrum of the sucrose density gradient to show the regions isolated for the experiments.

Added for a representative replicate gradient in the new figure, Figure 3-figure supplement 1.

**Reviewer #3 (Public Review):**
We thank the referee for his/her positive assessment of our study.Weaknesses:While no role of eIF2A in translation initiation is apparent, the authors do not determine what function eIF2A does play in yeast. Whether it plays a role in regulating translation in a different stress response is not determined.

We agree that there are many additional possibilities to consider for functions of eIF2A in translation initiation, including different stress situations or mutant backgrounds; however, we regard this as a limitation rather than a weakness in the experimental design and data obtained in the current study in which we examined the most likely possibilities for eIF2A function in yeast based on studies of the mammalian factor.

**Reviewer #3 (Recommendations For The Authors):**
Curiously, the authors indicate that they could not replicate published results for eIF2A's repressor function for URE2, PAB1, or GIC1 translation. This is a little concerning and one wonders if the yeast strain used in the previous study is different in some way from the authors' strain. Did the authors obtain that strain to test it in their assays?

The same WT and eIF2A∆ strains have been analyzed here and in the two cited studies on yeast IRESs.

The authors do discuss the fact that eIF2A may function to regulate translation in response to different stresses. It would have been a strength to test an alternative stress in the current study. However, I also appreciate that this could be the subject of a future study.

Agreed.

One minor question I have is whether the yeast strains used possess L-A dsRNA virus? While it may not be that this virus would necessarily mask a role of eIF2A-dependent translation, do the authors have any specific thoughts on this? Would different results be obtained if cured strains were used?

According to Ravoityte et al. (doi: 10.3390/jof8040381), the *S. cerevisiae* strain we employed, BY4741, harbors L-A-1 dsRNA; however, we have not explored whether curing the virus would alter the consequences of eliminating eIF2A.